# 3D in situ imaging of the female reproductive tract reveals molecular signatures of fertilizing spermatozoa in mice

**Lukas Ded[1,2], Jae Yeon Hwang[1], Kiyoshi Miki[3], Huanan F Shi[1], Jean-Ju Chung[1,4]***

[1]Department of Cellular & Molecular Physiology, Yale School of Medicine, New Haven, United States; [2]Laboratory of Reproductive Biology, Institute of Biotechnology, Czech Academy of Sciences, BIOCEV, Vestec, Czech Republic; [3]Boston Children's Hospital, Boston, United States; [4]Department of Obstetrics, Gynecology, and Reproductive Sciences, Yale School of Medicine, New Haven, United States

**Abstract** Out of millions of ejaculated sperm, a few reach the fertilization site in mammals. Flagellar $Ca^{2+}$ signaling nanodomains, organized by multi-subunit CatSper calcium channel complexes, are pivotal for sperm migration in the female tract, implicating CatSper-dependent mechanisms in sperm selection. Here using biochemical and pharmacological studies, we demonstrate that CatSper1 is an O-linked glycosylated protein, undergoing capacitation-induced processing dependent on $Ca^{2+}$ and phosphorylation cascades. CatSper1 processing correlates with protein tyrosine phosphorylation (pY) development in sperm cells capacitated in vitro and in vivo. Using 3D in situ molecular imaging and ANN-based automatic detection of sperm distributed along the cleared female tract, we demonstrate that spermatozoa past the utero-tubal junction possess the intact CatSper1 signals. Together, we reveal that fertilizing mouse spermatozoa in situ are characterized by intact CatSper channel, lack of pY, and reacted acrosomes. These findings provide molecular insight into sperm selection for successful fertilization in the female reproductive tract.

**\*For correspondence:**
jean-ju.chung@yale.edu

**Competing interests:** The authors declare that no competing interests exist.

## Introduction

In most mammals, millions or billions of spermatozoa are deposited into the cervix upon coitus. Yet less than 100 spermatozoa are found at the fertilization site, called the ampulla, and only 10–12 spermatozoa are observed around an oocyte (*Kölle, 2015*; *Suarez, 2002*). This implies the presence of mechanisms to select spermatozoa as they travel through the female reproductive tract and to eliminate non-fertilizing, surplus spermatozoa once the egg is fertilized (*Sakkas et al., 2015*). Recent ex vivo imaging studies combined with mouse genetics have shown that surface molecules on the sperm plasma membranes such as ADAM family proteins are essential for the sperm to pass through the utero-tubal junction (UTJ) (*Fujihara et al., 2018*). By contrast, whether such selection and elimination within the oviduct requires specific molecular signatures and cellular signaling of spermatozoa is not fully understood.

Mammalian sperm undergo capacitation, a physiological process to obtain the ability to fertilize the egg, naturally inside the oviduct (*Austin, 1951*; *Chang, 1951*). The emulation of sperm capacitation in vitro led to the development of in vitro fertilization (IVF) techniques (*Steptoe and Edwards, 1976*; *Wang and Sauer, 2006*). Since then, most studies on sperm capacitation and gamete interaction have been carried out under in vitro conditions. However, mounting evidence suggests that in vitro sperm capacitation does not precisely reproduce the time- and space-dependent in vivo events

**eLife digest** When mammals mate, males ejaculate millions of sperm cells into the females' reproductive tract. But as the sperm travel up the tract, only a handful of the 'fittest' sperm will actually manage to reach the egg. This process of elimination prevents the egg from being fertilized by multiple sperm cells and stops the eggs from being fertilized outside of the womb. A lot of what is known about fertilization in mammals has come from studying how sperm and eggs cells interact in a Petri dish. However, this approach cannot explain how sperm are selected and removed as they journey towards the egg.

Previous work suggests that a calcium channel, which sits in the membrane surrounding the sperm tail, may provide some answers. The core of this channel, known as CatSper, is made up of four proteins arranged into a unique pattern similar to racing stripes. Without this specific arrangement, sperm cells cannot move forward and fertilize the egg in time. To investigate the role of this protein in more depth, Ded et al. established a new way to image the minute structures of sperm cells, such as CatSper, in the reproductive tract of female mice.

Experiments in a Petri dish revealed that sperm cells that have been primed to fertilize the egg are a diverse population: in some cells one of the proteins that make up the calcium channel, known as CatSper1, is cleaved, while in other cells this protein remains intact. Visualizing this protein in the female reproductive tract showed that sperm cells close to the site of fertilization contain non-cleaved CatSper1. Whereas sperm cells further away from the egg – and thus closer to the uterus – are more likely to contain broken down CatSper1.

Taken together, these findings suggest that the state of the CatSper1 protein may be used to select sperm that are most likely to reach and fertilize the egg. Future studies should address what happens to the calcium channel once the CatSper1 protein is cleaved, and how this channel controls the movements and lifespan of sperm. This could help identify new targets for contraception and improve current strategies for assisted reproduction.

in the oviduct. Protein tyrosine phosphorylation (pY), which has been utilized as a hallmark of sperm capacitation over decades, showed different patterns in boar sperm capacitated in vitro from ex vivo and in vivo (*Luño et al., 2013*). In mice, pY is not required for sperm hyperactivation or fertility (*Alvau et al., 2016*; *Tateno et al., 2013*). Previous in vitro studies that represent the population average at a given time may or may not have observed molecular details of a small number of the most fertilizing sperm cells.

Capacitation involves extensive sperm remodeling that triggers cellular signaling cascades. Cholesterol shedding and protein modifications occur within the plasma membrane (*Visconti et al., 1999*; *Vyklicka and Lishko, 2020*). Cleavage and/or degradation of intracellular proteins by individual proteases and the ubiquitin-proteasome system (UPS) also participate in the capacitation process (*Honda et al., 2002*; *Kerns et al., 2016*). Various capacitation-associated cellular signaling pathways that include cAMP/PKA activation followed by pY increase and rise in intracellular pH and calcium result in physiological outcomes such as acrosome reaction and motility changes (*Balbach et al., 2018*; *Puga Molina et al., 2018*). The sperm-specific CatSper $Ca^{2+}$ channel forms multi-linear nanodomains on the flagellar membrane, functioning as a signaling hub that links these events and motility regulation during capacitation (*Chung et al., 2014*). Sperm from mice lacking the CatSper genes are unable to control pY development and fail to migrate past the UTJ (*Chung et al., 2014*; *Ho et al., 2009*). The presence and integrity of CatSper nanodomains, probed by CatSper1, correlate with sperm ability to develop hyperactivated motility (*Chung et al., 2017*; *Chung et al., 2014*; *Hwang et al., 2019*). It is not known how these molecular and functional events are coordinated in the individual sperm cells within the physiological context. The central hypothesis of this study is that specific modifications and processing of CatSper1, a pore subunit of CatSper channel, govern the channel biogenesis and $Ca^{2+}$ signaling state, endowing differential sperm motility and fertility during the fertilization journey in the female reproductive tract.

Here we reveal that most fertilizing mouse spermatozoa in situ are molecularly and functionally characterized by intact CatSper channel, lack of pY, and reacted acrosomes. Using biochemical and pharmacological analyses, we show that CatSper1 undergoes O-linked glycosylation during sperm

differentiation and maturation. Capacitation induces CatSper1 cleavage and degradation dependent on $Ca^{2+}$ influx and protein phosphorylation cascades. We find that CatSper1 processing correlates with pY development in the flagella among heterogenous sperm cells capacitated in vitro and in vivo. We use ex vivo imaging and microdissection to show that the intact CatSper channel is indispensable for sperm to successfully reach the ampulla and for the acrosome to react. Finally, we use newly developed 3D in situ molecular imaging strategies and ANN approach to determine and quantify the molecular characteristics of sperm distributed along the female reproductive tract. We demonstrate that spermatozoa past the UTJ are recognized by intact CatSper1 signals which are graded along the oviduct. These findings provide molecular insight into dynamic regulation of $Ca^{2+}$ signaling in the selection, maintenance of the fertilizing capacity, and elimination of sperm in the female reproductive tract.

## Results

### CatSper1 undergoes post-translational modifications during sperm development and maturation

We previously found that the CatSper channel complex is compartmentalized within the flagellar membrane, creating linear $Ca^{2+}$ signaling nanodomains along the sperm tail (*Chung et al., 2017*; *Chung et al., 2014*; *Hwang et al., 2019*). Caveolin-1, a scaffolding protein in cholesterol-rich microdomains, colocalizes with the CatSper channel complex but does not scaffold the nanodomain (*Chung et al., 2014*). The molecular weight and amount of intact CatSper1, among all CatSper subunits, specifically declines during sperm capacitation (*Chung et al., 2014*; *Figure 1B,D,H,I,K*). To better understand the molecular basis and functional implications of the unique processing of CatSper1, we first examined CatSper1 protein expression in the testis and epididymis. Interestingly, the molecular weight of CatSper1 increases gradually during sperm development and epidydimal maturation (*Figure 1A*; *top*), indicating that CatSper1 undergoes post-translational modifications. Next, we examined the nature of the modifications. Blocking tyrosine phosphatases by sodium orthovanadate or adding specific protein phosphatases, such as protein serine/threonine phosphatase 1(PP1) or protein tyrosine phosphatase 1B (PTP1B) to solubilized sperm membrane fraction, does not change the molecular weight of CatSper1 (*Figure 1—figure supplement 1A*). By contrast, when the sperm membrane was subjected to enzymatic deglycosylation, O-glycosidase, but not PNGase F, shifts the apparent molecular weight of CatSper1 to closest to the CatSper1 band that corresponds to the smallest molecular weight observed in the testis (*Figure 1A,B*, *Figure 1—figure supplement 1B*). These data suggest that CatSper1 in sperm is not a phosphoprotein but an O-linked glycosylated protein.

### CatSper1 resides in the subdomains of lipid rafts in mature sperm and is processed during capacitation

During sperm capacitation, cholesterol depletion destabilizes the plasma membrane and reorganizes the lipid raft (*Nixon et al., 2007*). One simple hypothesis is that the capacitation-associated changes in raft stability and distribution render CatSper1 accessible to protease activity. To test whether CatSper nanodomains are raft-associated, we performed sucrose density gradient centrifugation, which identified CatSper1 in lipid raft subdomains in mature sperm (*Figure 1C*). Before inducing capacitation, CatSper1 is not processed in sperm cells, probably because the CatSper1-targeting protease activity is normally not in the immediate vicinity to the CatSper nanodomains in the flagellar membrane (*Figure 1—figure supplement 1C,F*). Supporting this notion, the protease activity readily cleaves CatSper1 by solubilizing the sperm membrane fraction with Triton X-100 (*Figure 1—figure supplement 1C*).

### The CatSper1 N-terminus undergoes capacitation-associated degradation in vitro

Next, we investigated the location of CatSper1 cleavage and degradation using recombinant CatSper1 proteins and sperm lysates. The CatSper1 antibody used in this study is raised against the first N-terminal 150 amino acids of recombinant CatSper1 (*Ren et al., 2001*). C-terminal HA-tagged full-length (FL) or N-terminal deleted (ND) recombinant CatSper1 are expressed in HEK 293 T cells for

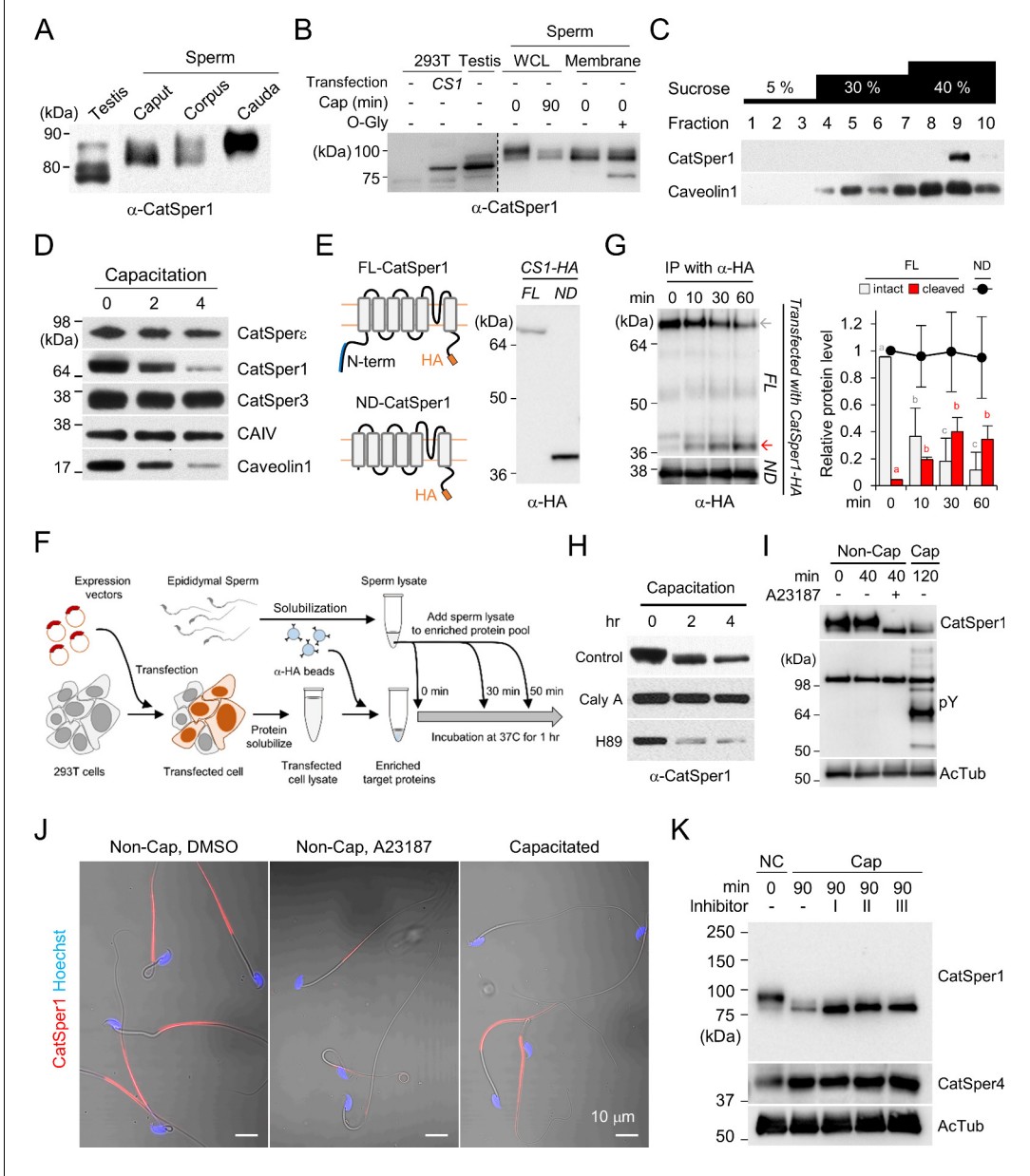

**Figure 1.** CatSper1 is specifically processed during in vitro capacitation. (**A–B**) CatSper1 undergoes post-translational modification during spermiogenesis and epididymal maturation. (**A**) A gradual decrease in electrophoretic mobility of CatSper1 is observed by western blot analysis. (**B**) Mouse CatSper1 is an O-glycosylated protein. Apparent molecular weights of CatSper1 proteins were analyzed by immunoblotting recombinant CatSper1 expressed in 293 T cells (*CS1*) and native CatSper1 from whole cell lysate (WCL) of testis and sperm (non-capacitated, 0 min; capacitated, 90 min) compared with those of CatSper in sperm membrane fraction treated with or without O-glycosidase (O-Gly). The dotted line indicates different exposure time of the same membrane. (**C**) CatSper resides in lipid rafts subdomains of the plasma membrane in mature sperm. Solubilized sperm proteins were fractionized by discontinuous sucrose density gradient (5, 30, and 40%) centrifugation. (**D**) CatSper1 is degraded during the late stage of capacitation. Protein expression levels of CatSper1 and caveolin-1, but not CatSper3, CatSperε, or carbonic anhydrase 4 (CAIV) are altered by in vitro capacitation. (**E–G**) CatSper1 is cleaved within the N-terminal domain (NTD). (**E**) A cartoon of full-length (FL, *top*) and N-terminal truncated (ND, *bottom*) recombinant CatSper1 protein expressed in the study (*left*). Both proteins are tagged with HA at their respective C-termini (orange). The CatSper1 antibody used in this study is raised against the 1–150 aa region of CatSper1 (blue, *Ren et al., 2001*). Detection of recombinant FL-CatSper1 and ND-CatSper1 expressed in 293 T cells (*right*). (**F**) A cartoon of the experimental scheme to test NTD truncation of CatSper1. FL-CatSper1 and ND-CatSper1 expressed in 293 T cells were solubilized and pulled-down using agarose resin conjugated with HA antibody. The enriched recombinant proteins were incubated with solubilized sperm lysates at 37°C for 0, 10, 30, and 60 min and subjected to immunoblot. (**G**) FL-CatSper1 is cleaved at NTD by solubilized sperm lysate. FL-CatSper1 (gray arrow) decreases while truncated form (red arrow) increases by incubation with solubilized sperm lysates (*top*). ND-CatSper1 proteins remain largely unchanged under the same conditions (*bottom*). The right panel shows

*Figure 1 continued on next page*

*Figure 1 continued*

quantification of the protein levels by measuring the band intensity of target proteins at each time point, normalized by total FL-CatSper1 at 0 min points (n = 3; intact, gray bars; cleaved, red bars) or ND-CatSper1 (n = 4; black dots). The sum of intact and cleaved FL-CatSper1 levels was used for total FL-CatSper1. Statistical analyses were performed between relative levels at each time point within each protein. Different letters indicate the significant difference. Data is represented as mean ± SEM. See also *Figure 1—source data 1*. Immunoblotting was performed with the HA antibody (**E** and **G**). (**H**) Capacitation-associated CatSper1 degradation is regulated by phosphorylation. CatSper1 degradation is accelerated by PKA inhibition. A PKA inhibitor, H89 (50 µM), enhances capacitation-associated CatSper1 degradation. A protein phophatase1 inhibitor, calyculin A (Caly A, 0.1 µM), prevents the CatSper1 degradation during sperm capacitation in vitro. (**I–J**) Ca²⁺ influx accelerates CatSper1 degradation. (**I**) CatSper activation during capacitation and Ca²⁺ ionophore treatment (A23187, 10 µM) facilitates the CatSper1 cleavage. (**J**) Representative immunofluorescence images of CatSper1 in the spermatozoa incubated under the conditions used in (**I**). The extent of CatSper1 degradation is heterogeneous in the capacitated sperm cells (*bottom*) compared with A23187-treated uncapacitated sperm cells (middle). (**K**) Capacitation-associated CatSper1 degradation is blocked by calpain inhibitors (I, II, and III). 20 µM of each calpain inhibitor was treated with sperm during capacitation. Blots shown here are representative of three independent experiments.

The online version of this article includes the following source data and figure supplement(s) for figure 1:

**Source data 1.** Relative amount of recombinant CatSper1 during in vitro proteolysis.
**Figure supplement 1.** Molecular mechanism for CatSper1 down-regulation during sperm capacitation.

pull-down and detection by western blot (*Figure 1E,F*). Solubilized sperm lysates degrade FL-CatSper1 and result in increased detection of cleaved CatSper1 by HA antibody (*Figure 1G*). By contrast, protein levels of recombinant ND-CatSper1 are not affected by incubation with sperm lysate (*Figure 1G*). These results demonstrate that the cytoplasmic N-terminal domain of CatSper1 is the target region for proteolytic activity in sperm cells. How is the CatSper proteolytic activity regulated?

## CatSper1 degradation involves Ca²⁺ and phosphorylation-dependent protease activity

At the molecular level, capacitation is initiated by $HCO_3$ uptake, which activates soluble adenylyl cyclase (sAC), resulting in increased cAMP levels. $HCO_3$ also stimulates CatSper-mediated Ca²⁺ entry into sperm cells by raising intracellular pH (*Kirichok et al., 2006*; *Figure 1—figure supplement 1F*). We thus examined whether the proteolytic activity requires cAMP/PKA and/or Ca²⁺ signaling pathways. Adding a PKA inhibitor H89 or the St-Ht31 peptide, which abolishes PKA anchoring to AKAP, accelerated CatSper1 degradation during sperm capacitation (*Figure 1H*, *Figure 1—figure supplement 1D*). Consistently, calyculin A, a serine/threonine protein phosphatase inhibitor, suppresses the capacitation-associated CatSper1 degradation (*Figure 1H*). These data suggest that regulation of the proteolytic activity targeting CatSper1 involves protein phosphorylation cascades. Interestingly, adding Ca²⁺ ionophore A23187 to the sperm suspension induces CatSper1 processing even under non-capacitating conditions that either do not support PKA activation or outright inhibit PKA activity (*Figure 1I,J*, *Figure 1—figure supplement 1E*, *top*). This effect of Ca²⁺ influx by A23187 is not simply due to a rise in the intracellular Ca²⁺ but presumably also requires membrane events because loading sperm with BAPTA-AM cannot prevent the proteolytic activity under capacitating conditions (*Figure 1—figure supplement 1E*, bottom). Thus, we hypothesize a Ca²⁺ dependent protease that is indirectly regulated by protein phosphorylation such as calpain (*Ono et al., 2016*) might process CatSper1. We observed that all three classes of calpain inhibitors prevent CatSper1 from capacitation-associated degradation (*Figure 1K*). All these results corroborate our hypothesis that the responsible protease(s) is associated with the flagellar membrane but is not localized inside the CatSper nanodomains (*Figure 1—figure supplement 1F*); capacitation-associated membrane reorganization and the increase in local Ca²⁺ via the CatSper channel activates the protease(s). We speculate that this pathway can be indirectly modulated by PKA such as via regulation of the protease activity by PKA phosphorylation of PP1/PP2A.

## CatSper1 degradation correlates with pY development in sperm cells capacitated in vitro

Inducing sperm capacitation in vitro results in a functionally heterogeneous sperm population in which no more than ~15% of cells are hyperactivated (*Neill and Olds-Clarke, 1987*). This is because individual sperm cells undergo time-dependent changes. Accordingly, the extent to which CatSper1

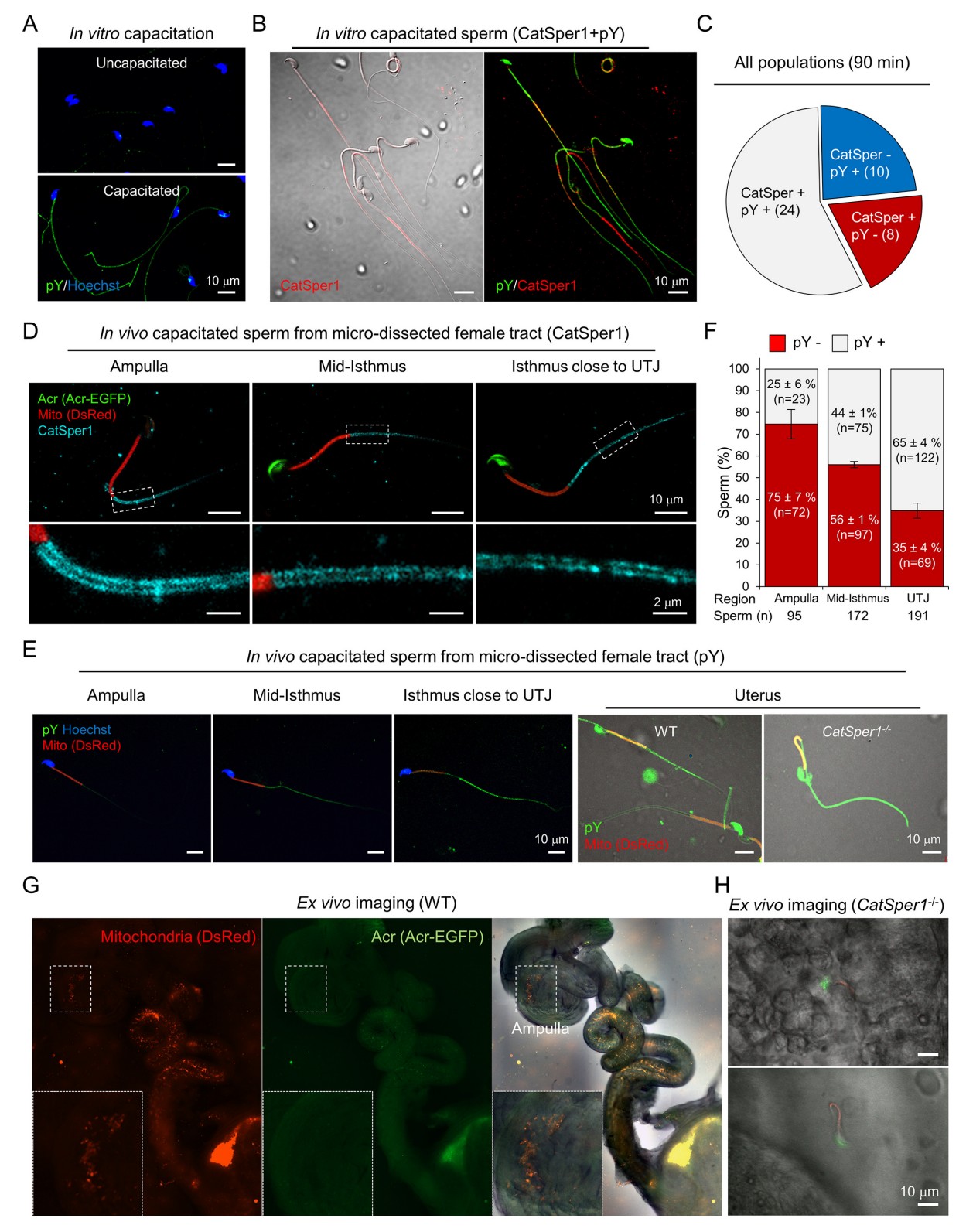

**Figure 2.** Sperm cells become heterogeneous functionally and molecularly along the female tract. (**A**) Immunodetection of pY from in vitro capacitated sperm. (**B**) Sperm cells that maintain intact CatSper1 during in vitro capacitation exhibit reduced pY development. A representative image of CatSper1 (red) is merged with the corresponding DIC image (left) or pY image (right). Sperm were capacitated in vitro for 90 min (**A–B**). (**C**) A representative pie chart showing expression patterns of CatSper1 and pY in individual sperm cells of a single population of in vitro capacitated sperm. Sperm number in

*Figure 2 continued on next page*

*Figure 2 continued*

each group are indicated in parentheses. (**D–F**) Sperm cells capacitated in vivo show distinct molecular characteristics along the female tract. The degrees of CatSper1 processing (**D**) and development of tyrosine phosphorylation (pY) (**E–F**) during in vivo capacitation were analyzed by immunostaining of the sperm cells at different regions of microdissected female tracts 8 hr post-coitus. The indicated regions are magnified to show distributions of CatSper1 in sperm cells. Sperm cells that arrived at the ampulla are acrosome reacted (Acr-EGFP-negative) and CatSper1 intact (**D**) and lack pY development (**E–F**). Gradual increase of pY is observed in the oviductal sperm located closer to UTJ (**E–F**). Sperm cells that fail to pass UTJ and reside in the uterus show heterogeneous patterns of pY. *Catsper1*^-/-^ sperm recovered from the uterus of a mated female show robust elevation of pY. Red, DsRed in mitochondria (Mito). (**F**) Sperm cells from each region of the female tract were counted and classified for pY positive (gray) or pY negative (red) from ampulla (n = 8), mid-isthmus and UTJ, respectively (n = 9) from 5 females at 8 hr post-coitus. Data is represented as mean ± SEM. See also *Figure 2—source data 1*. (**G–H**) WT sperm cells, but not *Catsper1*^-/-^ sperm cells, which arrive at the ampulla are acrosome reacted. Ex vivo imaging of female tracts mated with WT (**G**) and *Catsper1*^-/-^ males (8 hr post-coitus). (**G**) WT sperm cells are acrosome-reacted at the ampulla (EGFP-negative, *inset*). (**H**) A few *Catsper1*^-/-^ sperm cells observed at ampulla have intact acrosome (EGFP-positive). Red, DsRed in mitochondria; green, EGFP tagged to Acr (Acr-EGFP); Merged, fluorescent images merged with the corresponding DIC image. Fluorescence micrographs shown here are representative of at least three independent experiments.Su9-DsRed2/Acr-EGFP WT and Su9-DsRed2/Acr-EGFP *Catsper1*^-/-^ mice (*Chung et al., 2014*) were used for mating (D, E, G, and H).

The online version of this article includes the following source data for figure 2:

**Source data 1.** Classification and quantification of pY from in vivo capacitated sperm.

---

degrades varies with individual sperm cells capacitated in vitro (*Figure 1J*, *Figure 2A–C*). The presence and integrity of the CatSper nanodomains, probed by the CatSper1 antibody, is an indicator of sperm capability to hyperactivate (*Chung et al., 2017*; *Chung et al., 2014*). Next, we examined the functional relevance of pY to sperm that can hyperactivate. Notably, we find that sperm cells that maintain intact CatSper1 develop capacitation-associated pY to a lesser degree in vitro (*Figure 2B, C*). This finding is consistent with the reported phenotype of *Catsper1* knockout sperm that exhibit potentiated pY during capacitation (*Chung et al., 2014*). Thus far, our results suggest that in vitro capacitation generates a heterogeneous sperm population in which intact CatSper1 and pY development are inversely correlated at the single cell level. These heterogeneous sperm cells in vitro might reflect a collection of the time- and space-dependent changes that sperm undergo in the oviduct (*Chang and Suarez, 2012*; *Demott and Suarez, 1992*).

## Sperm cells capacitated in vivo become heterogeneous along the female tract with distinct molecular characteristics

To assess molecular changes of CatSper1 and pY in the context of sperm capacitation in vivo, we utilized Su9-DsRed2/Acr-EGFP male mice (*Hasuwa et al., 2010*). Sperm from these transgenic mice have green acrosomes (EGFP) and red mitochondria (DsRed2); loss of GFP indicates a reacted acrosome and RFP allows detection of sperm regardless of the acrosome state. We performed microdissection to obtain the spatially distributed sperm populations along the female reproductive tract mated at 8 hr post-coitus and flushed out sperm cells from different regions. By subsequent immunostaining, we found that CatSper1 in the spermatozoa that passed the utero-tubal junction (UTJ) are arranged normally along the tail, mostly protected from degradation, but in decreasing intensity and continuity more towards UTJ (*Figure 2D*). In striking contrast, pY is not readily detected in the spermatozoa from the ampulla but appears in the oviductal sperm increasingly towards UTJ (*Figure 2E,F*). Absence of EGFP reveals that spermatozoa from the ampulla are fully capacitated and acrosome reacted (AR) but those in the isthmus are undergoing AR (*Figure 2D,G*). Ex vivo imaging of Su9-DsRed2/Acr-EGFP sperm in the reproductive tract removed from mated female mice reveals segment-specific patterns of the acrosome status (*Figure 2G*). This result is consistent with the previous observations that AR initiates in the mid-isthmus (*Hino et al., 2016*; *Muro et al., 2016*) and reacted spermatozoa are able to penetrate the zona in vivo (*Jin et al., 2011*). Interestingly, we found that a few *Catsper1*^-/-^ sperm cells that managed to arrive at the ampulla are all not acrosome reacted (*Figure 2H*), supporting the notion that CatSper-mediated Ca$^{2+}$ signaling is required for sperm acrosome reaction (*Stival et al., 2018*). These results suggest that escape of CatSper1 from the cleavage and subsequent degradation suppresses pY development, enabling sperm to maintain hyperactivation capability, prime AR, and achieve the fertilization in vivo.

## 3D in situ molecular imaging of gametes in the female reproductive tract

The physiological importance of tracing a small number of spermatozoa progressing to the fertilization site prompted us to seek a method that enables direct molecular assessment of single cells inside the intact female tract. We have adapted tissue clearing technologies to establish three-dimensional (3D) in situ molecular imaging systems for fertilization studies (*Figure 3*, *Figure 3—figure supplement 1*, *Figure 3—videos 1–6*). We found that various tissue clearing methods (*Chung et al., 2013*; *Murray et al., 2015*; *Yang et al., 2014*) are applicable to the reproductive organs from both male and female mice to preserve gross morphology, and fine cellular and subcellular structures. The cleared tissues preserved protein-based fluorescence and were compatible with labeling with dyes and antibodies; growing follicles (WGA) inside the ovary (phalloidin), oviductal folds (WGA) and multi-ciliated (anti-Ac-Tub) epithelium (PNA), different stages of male germs cells (Acr-EGFP) in the seminiferous tubules of the testis and the epididymis are readily detected (*Figure 3A–C*, *Figure 3—figure supplement 1*, *Figure 3—videos 1–6*). 3D volume imaging of the whole cleared female tract labeled by WGA well illustrates the uterine and isthmic mucus and the labyrinths of passages sperm must navigate (*Figure 3A,B*, *Figure 3—figure supplement 1B,C*, *Figure 3—videos 3*, *5*, *6*). Moreover, 3D rendering of the images and digital reconstruction of oviductal surface and central lumen depicts continuous and non-disrupted morphology (*Figure 3D,E*) consistent with reported dimensions and parameters (*Stewart and Behringer, 2012*), validating the integrity of the processed oviduct.

Next, we combined tissue clearing with an in vivo sperm migration assay (*Chung et al., 2014*; *Yamaguchi et al., 2009*) to molecularly analyze different sperm populations during the fertilization process. Among tested clearing methods, we found that passive clearing of CLARITY-processed reproductive tracts from time-mated females retains the location and stability of gametes within the track past UTJ (*Figure 3F–I*, *Figure 3—videos 4*, *7* and *Figure 4—video 1*); whole-animal fixation by trans-cardiac perfusion perturbs minimally and rapidly arrests all cellular function while tissue-hydrogel matrix fills the lumen of the oviduct and provides supportive meshwork to prevent gamete loss during subsequent labeling steps (*Figure 3—figure supplement 2*). This new in situ imaging platform enables capturing a moment of sperm-egg interaction; a spermatozoon that approaches a fertilized egg protruding the 2nd polar body in the ampulla is detected in a cleared female tract 8 hr post-coitus immunostained by acetylated tubulin antibody (Figure 3, *Figure 3—video 7*). CatSper1 antibody specifically recognizes sperm cells transfixed inside the ampulla in cleared female tract (*Figure 3H,I*). Tissue clearing allows 3D volume imaging of the female tract but does not compromise the resolution. Two linear CatSper1 domains typically observed by confocal imaging are easily observed in the sperm cells inside an ampullar region of the whole cleared female tract (*Figure 3I*). Thus, the integrity of CatSper1 in sperm cells at different locations along the female tract can be subjected to quantitative analysis.

## Sperm cell that successfully reach the ampulla are CatSper1-intact and acrosome reacted

With this new imaging strategy to detect sperm cells that remain transfixed in the female reproductive tract (*Figure 3*), we investigated acrosome state and CatSper1 integrity in sperm populations directly from the cleared tract of females 8 hr after mating, focusing on a few anatomically defined regions (*Figure 4*). Based on the earlier results from micro-dissection or ex vivo imaging (*Figure 2D–H*), we anticipated that sperm cells that successfully reach the ampulla would be CatSper1-intact and acrosome reacted. As expected, most sperm cells located in the ampulla exhibit linearly arranged intact CatSper1 and reacted acrosomes (*Figure 4A*, *top*, *Figure 4—video 1*). In the middle isthmus, both CatSper1 and acrosome remain intact in most sperm cells, but mixed patterns are observed in some cells (*Figure 4A*, *middle*, *Figure 4—video 1*). Interestingly, acrosome is largely intact in the sperm clusters in the proximal isthmus close to UTJ whereas CatSper1 is barely detected (*Figure 4A*, *bottom*, *Figure 4—video 1*). This contrasts with the reduced but readily visible CatSper1 in the sperm from the same region by microdissection (*Figure 2D*). It is possible that the relatively longer tissue processing time and subsequent labeling could have contributed to lower the signal to noise ratio to a certain degree. Notably, 3D volume imaging of this mid isthmus regions reveals sperm cells aligned in one direction towards the ampulla, providing unprecedented insight

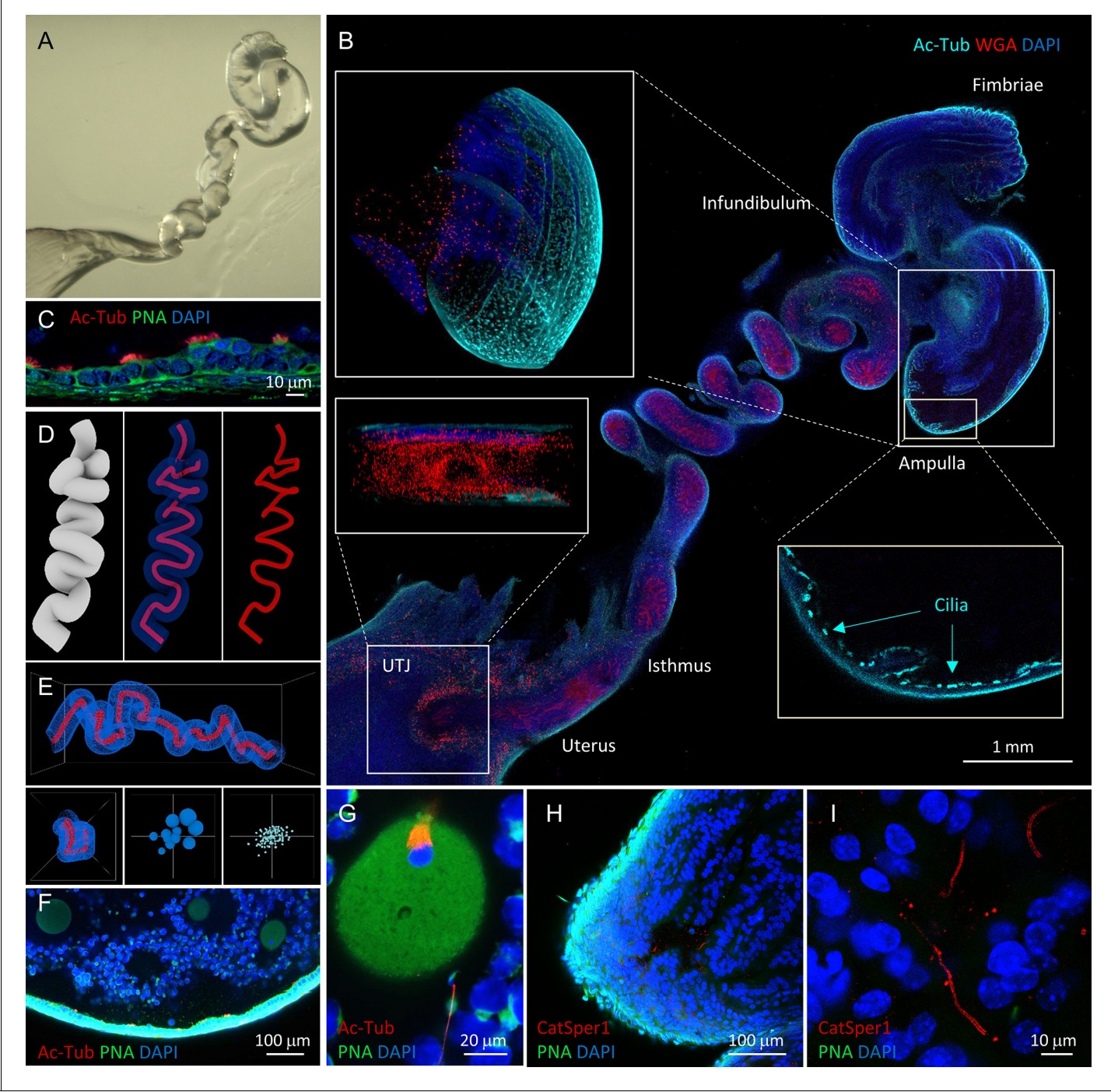

**Figure 3.** Tissue clearing preserves morphology of female reproductive tract and enables molecular imaging and post-processing of gametes in situ. (A) Refractive index-matched cleared mouse female reproductive tract by CLARITY-based tissue clearing. (B) Optical imaging of the cleared female reproductive tract stained by WGA (red), Ac-Tub antibody (cyan) and DAPI (blue), 100×. Insets show cilia stained by Ac-Tub antibody in 2D (*bottom right*), a 3D (*top left*) projection of the ampulla, and a UTJ cross-section (*bottom left*). (C) Details of the ampullar epithelium stained by PNA (green), Ac-Tub antibody (red) and DAPI (blue), 400×. (D) 3D digital image reconstruction of the oviduct representing different 3D images rendered for oviductal surface (*left*) and central lumen of oviduct with (*middle) or* without (*right*) oviductal volume information. (E) Morphometric and fluorescent signal quantification analysis of the oviduct showing the morphometric meshwork representation of the 3D volumetric data from the oviduct imaging (*top*), the corresponding side view (*bottom left*) and the non-numerical visual representations of the basic volumetric (*bottom middle*) and fluorescent (*bottom right*) properties. (F) A fluorescent image showing a closer look of the cleared ampulla with oocytes (oocyte magnified in the panel G on the right-most side), 100×. (G) An oocyte with the meiotic spindle; a sperm cell is approaching the *zona pellucida* directly inside the ampulla, PNA (green), anti-AcTub antibody (red) and DAPI (blue), 630×. (H) A tile-scanned confocal image of the epithelium of the cleared ampulla (8 hr post-coitus) stained by anti-

*Figure 3 continued on next page*

*Figure 3 continued*

CatSper1 antibody (red), PNA (green), and DAPI (blue), 100×. (I) Details of the sperm stained directly inside the ampulla by anti-CatSper1 antibody (red). Two linear CatSper domains are clearly recognizable by confocal imaging. Cell nuclei are stained with DAPI (blue); acrosomes are stained with PNA (green). Images shown here are representative of at least three independent experiments. See also *Figure 3—videos 3–7*.

The online version of this article includes the following video and figure supplement(s) for figure 3:

**Figure supplement 1.** Multicolor 3D fluorescence in situ imaging of cleared mouse reproductive organs.

**Figure supplement 2.** The effect of various components used in tissue clearing procedure on sperm cells.

**Figure 3—video 1.** 3D volume imaging of a whole ovary; related to *Figure 3*.

https://elifesciences.org/articles/62043#fig3video1

**Figure 3—video 2.** 3D Volume imaging of the testis and epididymis; related to *Figure 3*.

https://elifesciences.org/articles/62043#fig3video2

**Figure 3—video 3.** 3D rotational movie of whole female reproductive tract.

https://elifesciences.org/articles/62043#fig3video3

**Figure 3—video 4.** z-stack movie of the ampulla.

https://elifesciences.org/articles/62043#fig3video4

**Figure 3—video 5.** z-stack movie of the oviduct.

https://elifesciences.org/articles/62043#fig3video5

**Figure 3—video 6.** 3D rotational movie of the UTJ.

https://elifesciences.org/articles/62043#fig3video6

**Figure 3—video 7.** Moments after fertilization.

https://elifesciences.org/articles/62043#fig3video7

---

into sperm taxis in the fertilization process (*Figure 4—video 1*). Our qualitative but semi-quantitative analyses suggest that CatSper1 is largely protected from degradation once in the oviduct; acrosome reaction initiates in the mid-isthmus and is completed in the ampulla before interacting with the oocytes (*Figure 4B*). These results are consistent with our initial observations from microdissection and ex vivo imaging studies (*Figure 2D,G*), validating the information obtained by our in situ molecular imaging platform. Taken together, we conclude that intact CatSper1, lack of pY, and reacted acrosome are molecular and functional signatures of most fertilizing spermatozoa in the physiological context.

## Automatic detection of sperm in the voluminous female tract using artificial neural network

Processing 3D volumetric fluorescent data presents a significant challenge; analyses of sperm in the female tract includes object identification in the voluminous specimen, object separation from background noise, and object alignment in three dimensions. To address these logistics problems, we took an advantage of the artificial neural network (ANN) approach for automatic localization and signal isolation. We performed a proof-of-principle investigation utilizing CatSper1 distributions in sperm cells from our 3D in situ molecular imaging (*Figure 5*). First, we manually annotated 3D fluorescent signatures of sperm, somatic nuclei and background noise from the original images. These signatures were placed in different abundance models in the ANN 3D training environments (*Figure 5A*, *Figure 5—figure supplement 1*, *Figure 5—video 1*) for subsequent ANN training using MatLab ANN module. We performed a supervised iteration process where the sperm locations were predefined in the training environments (*Figure 5—figure supplement 2A*). We evaluated the performance of individual ANN according to their sensitivity and specificity in detecting sperm cells and somatic nuclei, and the abundance (voxel occupancy) of noise (*Figure 5B*, *Figure 5—figure supplement 2B,C*). Detection sensitivity is chosen as a major parameter used to evaluate the ANN performance in the training environment simulated with the values similar to those in real samples. The specificity required for sperm detection is lower than the sensitivity, thus provides mainly empty analytical frames that are easily removed manually. After iteration and performance evaluation, we selected the best performing ANN and analyzed images from our experimental samples for which we manually counted sperm number (*Figure 5C*). The selected ANN is able to recognize all the sperm detected manually and the ANN sensitivity varies around 90% in individual samples (*Figure 5—figure supplement 2C*), validating the ANN performance. Furthermore, the false-negative detection of sperm all comes from the sperm with dubious signals in the antecedent human eye

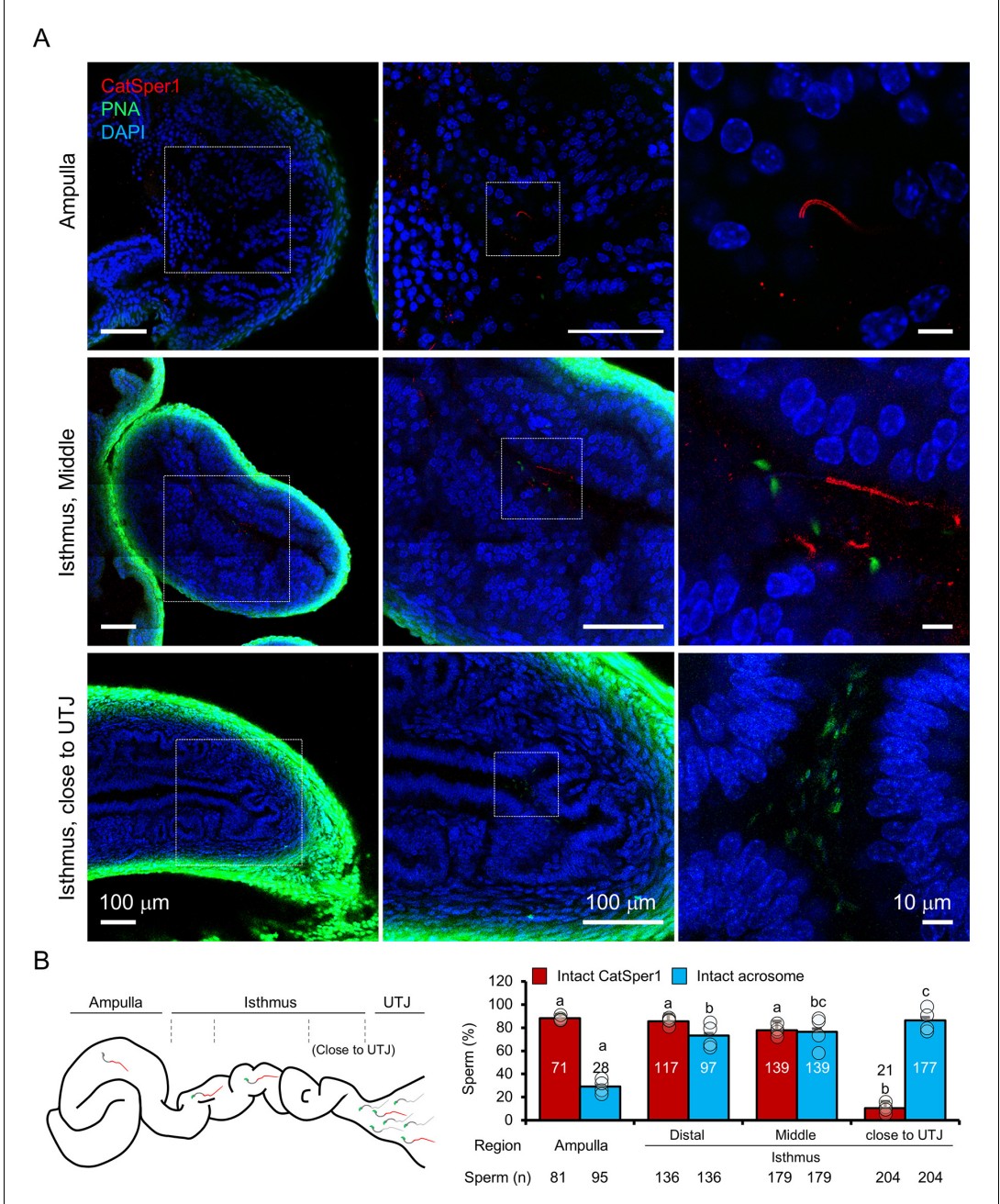

**Figure 4.** In situ molecular imaging of sperm reveals the changes in acrosomal status and CatSper1 fluorescent patterns during capacitation along the female reproductive tract. (**A**) Representative fluorescent confocal microscope images of acrosome and CatSper1 fluorescent patterns from three different regions along the cleared female reproductive tract (Ampulla, *top*; Middle isthmus, *middle*; Proximal isthmus, *bottom*) with different magnifications of the corresponding areas. (**B**) A cartoon image of the female reproductive tract showing the approximate boundaries between the regions of interest (*left*) used as grouping variable in the subsequent quantification (*right*). The total number of CatSper1-intact sperm (red columns) or acrosome-intact sperm (blue columns) counted from four independent experiments (n = 4) is shown (*bottom*). Circles indicate the proportion of sperm cells in each examined site from an independent experiment. Statistical analyses were carried out with one-way analysis of variance (ANOVA) with Tukey post hoc test. Means with different letters indicate significant difference (p<*0.05*) in pairwise comparison between the different regions of female tract. Data is represented to mean ± SEM. See also *Figure 4—source data 1* and *Figure 4—video 1*.

The online version of this article includes the following video and source data for figure 4:

**Source data 1.** Sperm counts for intact CatSper1 and acrosome from in situ imaging.

**Figure 4—video 1.** Sperm in the different regions of the oviduct.

https://elifesciences.org/articles/62043#fig4video1

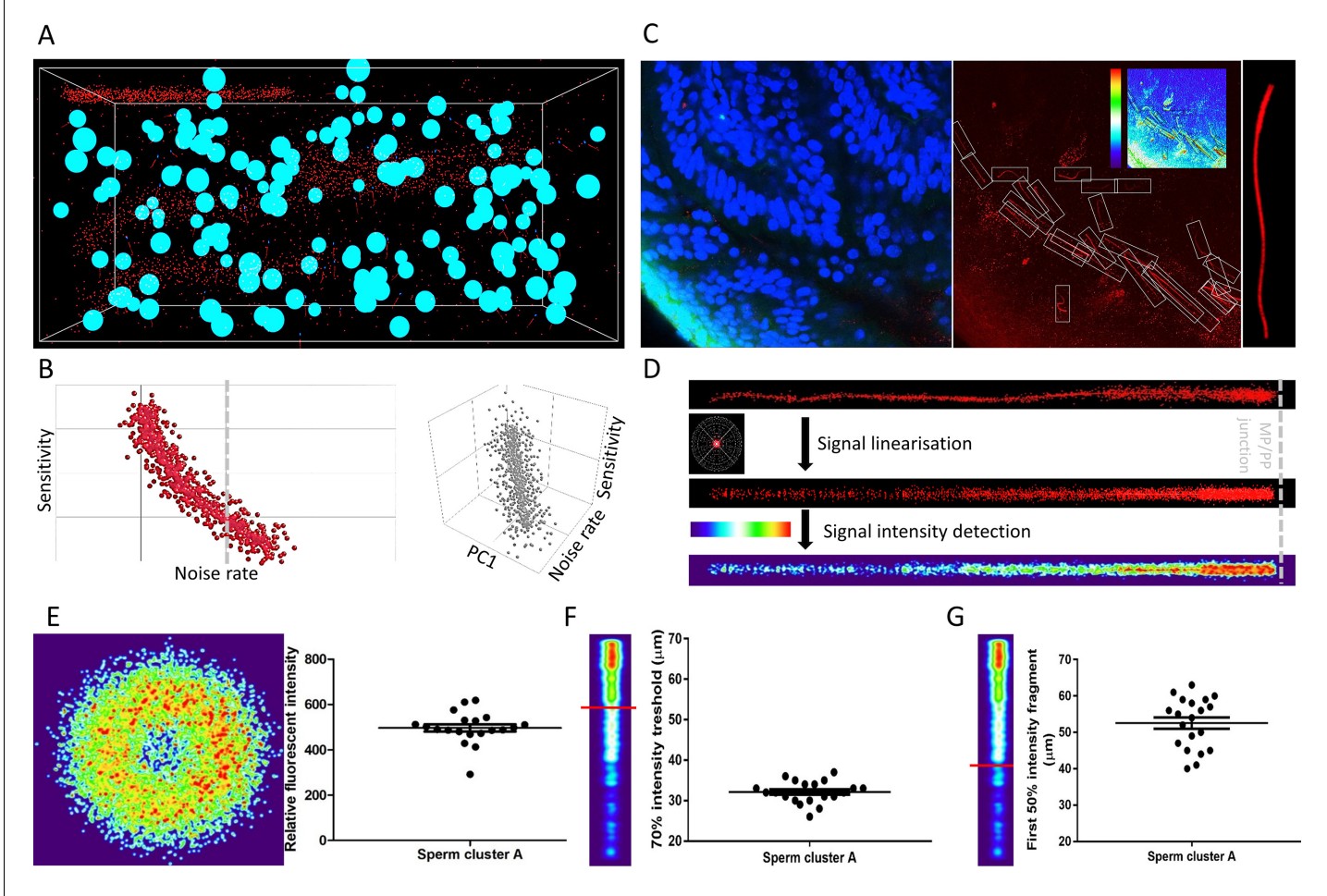

**Figure 5.** ANN automatically detects fluorescent patterns from 3D volume images of a cleared female reproductive tract, enabling isolation and statistical comparisons of sperm cells. (**A**) A 3D training environment for ANN emulating sperm cells, somatic nuclei, and noise. (**B**) Examples of ANN training statistics showing the trend of correlation between the noise rate in the training environment and the sensitivity of ANN (left), and a three-dimensional correlation trend between noise rate in the training environment, sensitivity and principal component (PC1) consisting of sperm and nuclei abundancies (right). $N_{sampling\ environment}$ = 1000, individual data points plotted. (**C**) A microscopic focal plane image of the sperm cluster inside the cleared female reproductive tract used for evaluating the ANN performance in real sample (*left*), its superposition in the CatSper1 channel with the individual sperm tails in detection frames with the inset analytical heatmap (middle) and the magnification of one of the analytical frames with a CatSper1-positive sperm tail (right). (**D**) Representation of the fluorescent signal in the sperm tail after normalizing individual voxels to signal from the corresponding sperm nucleus (top), after applying linearization and overlay algorithms (middle), and heatmap representation of the relative fluorescent intensities among multiple sperm tail (bottom). (**E**) Analysis of the relative intensities of the fluorescent signals from sperm located inside the mid-isthmus cleared female reproductive oviduct. The left panel represents the intensity of CatSper1 fluorescent signal in the cross-section of one sperm tail from 20 individual sperm under analysis (middle isthmus). The right panel show the distribution of relative fluorescent intensity of 20 sperm. (**F**) Analysis of the continuity of the fluorescent signal along the individual sperm tails; the first fragment of the 70% signal intensity decreases from the midpiece/principal piece interface. (**G**) The first fragment of the 50% signal intensity decreases from the midpiece/principal piece interface. Graphs in E-G represent Mean ± SEM. with individual measurements are plotted as dots (N = 20). See also *Figure 5—source data 1* and *Figure 5—video 1*.

The online version of this article includes the following video, source data, and figure supplement(s) for figure 5:

**Source data 1.** ANN development source materials.
**Figure supplement 1.** Generation of training environments for ANN development.
**Figure supplement 2.** A workflow diagram for ANN and performance evaluation.
**Figure supplement 3.** Image post-processing of sperm detected inside the cleared female reproductive tract.
**Figure 5—video 1.** A flight-through movie of analytical space without noise, related to *Figure 5A*.
https://elifesciences.org/articles/62043#fig5video1

evaluation; the 90% of the ANN detected sperm expresses well recognizable CatSper1 fluorescent staining patterns (*Figure 5C*, *Figure 5—figure supplement 3A*).

In order to pair each CatSper1 signal containing tail with the head from the same cell in the subsequent analysis, we took the reverse approach to the environment production by removing the detected noise and somatic cell nuclei from the analytical frames (*Figure 5—figure supplement 3B*). The pre-processed CatSper1 fluorescent signal were then subjected to subsequent alignment, pattern linearization, and intensity detection (*Figure 5D*, *Figure 5—figure supplement 3C*). These steps make possible calculation and visual representation of the fluorescent intensity parameters along the sperm tail related to their CatSper1 integrity status (*Figure 5E–G*).

## ANN-quantified CatSper1 signal reveals a molecular signature of successful sperm in situ

The quadrilateral and linear organization of the $Ca^{2+}$ signaling nanodomains discovered by super-resolution imaging of CatSper1 (*Figure 6A*) is an indicator of a sperm cell's ability to hyperactivate and fertilize the egg in vitro (*Chung et al., 2017*; *Chung et al., 2014*). The present study demonstrates that incubating sperm cell under capacitating conditions in vitro induces CatSper1 cleavage and degradation, leading to a heterogeneous sperm population (*Figures 1* and *2*). Building on our observations of sperm cells from microdissection, ex vivo imaging, and CLARITY-based in situ molecular imaging (*Figures 2*, *3* and *4*), we hypothesize that CatSper1 is a built-in countdown timer for sperm death and elimination in the female tract; CatSper1 cleavage and degradation, triggered in a time- and space-dependent manner along the female tract, signals to end sperm motility, and ultimately sets sperm lifetime in vivo. With our newly developed automated ANN method to obtain high-quality 3D fluorescent images of CatSper1 in the sperm cells from cleared female tract samples, we further tested this idea by quantitatively analyzing the CatSper1 signals in situ.

Our in situ imaging platform offers the typical resolution that a confocal microscopy can provide; two separated CatSper1 arrangement along the sperm tail (*Chung et al., 2017*) are detected without any computational processing (*Figures 3I* and *4A*). This encouraged us to develop an analytical procedure to assess the status of CatSper1 quadrilateral and linear distributions. We isolated the fluorescent signal from a proximal region of the principal piece close to the annulus where the immunolabeled CatSper1 signal is the most intense (*Figure 6B*). To superpose the individual cross-sectional images according to the expected four intensity peaks, we aligned randomly oriented transversal-projection images by placing the quadrant with the highest fluorescent intensity to top right corner (*Figure 6B*, *inset*). The aligned images were then superposed (*Figure 6C*) and used for statistical purposes to represent quadrilateral arrangement of CatSper1 in individual sperm cells (*Figure 6D*). The individually processed images of sperm cells from the oviductal regions close to UTJ, middle isthmus, and ampulla were again superposed to create cumulative diagrams and heat maps corresponding to these regions (*Figure 6E*). They show quadrilateral distribution of enriched CatSper1 signal more clearly from the sperm population in the ampulla compared to the population in the oviduct close to UTJ (*Figure 6E,G,H*).

To further quantify and statistically analyze our outputs, we divided the pre-processed images of individual sperm cells on 80 round areas (*Figure 6—figure supplement 1A*) and calculated fluorescent intensities among them. The quantified intensity from the 80 areas were plotted; the observed four peaks (highest intensity) and valleys (lowest intensity) were used to calculate the delta value among them to represent the quality of CatSper1 quadrilateral structure (*Figure 6F*). Our quantitative analysis (*Figure 6G*, *Figure 6—figure supplement 1B*) shows consistent results with our previous semi-quantitative analysis by manual assignment of the CatSper1 patterns (*Figure 4*). Together with the whole tissue image processing (*Figure 3E*), the quantitative analysis clearly visualizes that sperm populations located along the cleared oviduct have statistically different CatSper1 quadrilateral intensity delta values (*Figure 6H*).

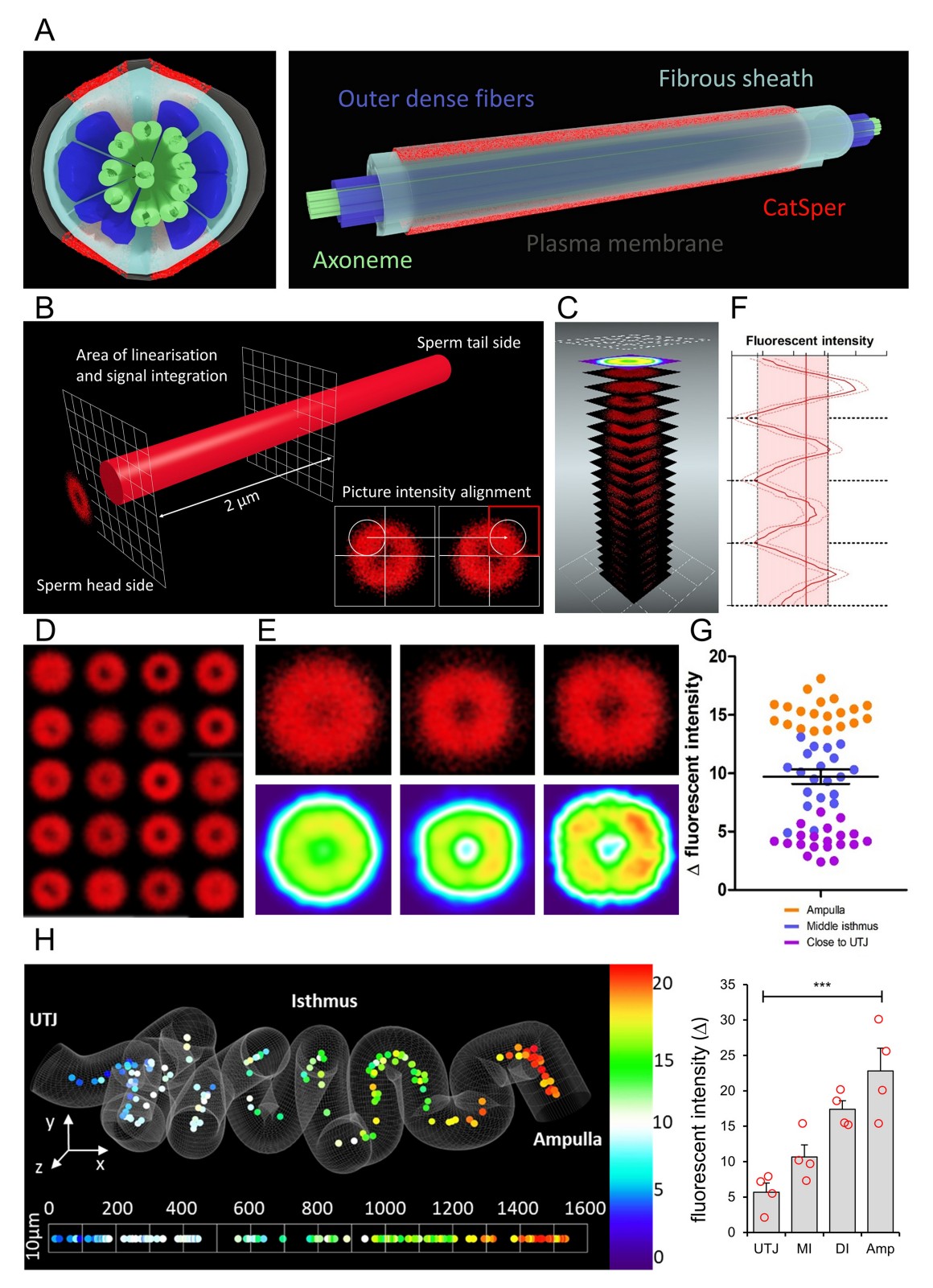

**Figure 6.** ANN assessment of quadrilateral CatSper nanodomains and Δ fluorescent intensity in sperm population along the cleared female tract immunolabeled for CatSper1 conforms to findings by other approaches used in this study. (**A**) 3D perspective schematic views of quadrilateral CatSper nanodomains. A cross-section (left). A side view (right). (**B**) A schematic diagram describing the image processing procedure. (**C**) An illustration of generating a heatmap from the pre-processed micrographs. (**D**) 20 processed micrographs of the CatSper1 signal from the sperm cluster from middle

*Figure 6 continued on next page*

*Figure 6 continued*

isthmus of the cleared oviduct. (**E**) Processed micrographs (top) and their corresponding heatmaps (bottom) from 20 spermatozoa from the oviduct close to UTJ (left), middle isthmus (middle), and ampulla (right). (**F**) An example of fluorescent intensity analysis of processed images showing the four peaks corresponding to four CatSper1 quadrilateral domains and calculated averaged Δ value (transversal red line). (**G**) Analysis of the fluorescent intensity differences (Δ values; red area in panel F) among three sperm populations from ampulla, middle isthmus, and isthmus close to UTJ. Graph represents Mean ± SEM. with individual measurements are plotted as dots (N = 60). (**H**) A topological heatmap showing the integrity of the quadrilateral CatSper domain organization represented by Δ values of the fluorescent intensity along the morphometrical space of the cleared oviduct (left, $N_{sperm}$ = 152) with the corresponding inferential statistical analysis of the differences of the signal intensities (Δ values, right) among four sperm populations (Amp – Ampulla, DI – Distal Isthmus close to ampulla, MI – middle isthmus, UTJ – utero-tubal junction). Bars denote Mean ± SEM. The baseline indicates homogenous angular distribution of the CatSper fluorescent signal with no quadrilateral distributions. Statistical significance was calculated using KW-ANOVA, (***p<0.001), $N_{animals}$ = 4. See also *Figure 6—source datas 1* and *2*.

The online version of this article includes the following source data and figure supplement(s) for figure 6:

**Source data 1.** Quantification of Δ values of CatSper1 fluorescent intensity.
**Source data 2.** 3D in situ analytical tools for CatSper quadrilateral structure.
**Figure supplement 1.** Analysis of the transversal quadrilateral CatSper1 domains organization in the tail of sperm inside the cleared female reproductive tract.

## Discussion

### CatSper1 as a molecular barcode for sperm maturation and transition in the female tract

Testicular spermatozoa undergo maturation and biochemical alterations in the intraluminal environment of the epididymis (*Cornwall, 2009*). Glycan-modifying enzymes such as glycosidases and glycosyltransferases are present in the epididymal luminal fluid (*Tulsiani, 2003*). Here we have shown that CatSper1 in mouse sperm is an O-linked glycosylated protein with gradually increasing molecular weight from the testis to the epididymis during male germ-cell development. The different forms of native CatSper1 may represent different degrees of glycosylation. Heterologously expressed CatSper1 cannot reach the plasma membrane, remaining instead at the ER/Golgi (*Chung et al., 2017*; *Chung et al., 2011*; *Ren et al., 2001*). It is intriguing that the molecular weight of recombinant CatSper1 is similar to one of the testicular forms of CatSper1 but bigger than that of the enzymatically deglycosylated and naked polypeptide. O-linked glycosylation takes place in the cis-Golgi for secreted and transmembrane proteins after the protein is folded (*Röttger et al., 1998*), suggesting that additional modification is required for native CatSper1 to exit the Golgi. Determining the precise identity and modification site may help to clarify the long-sought functional expression of the CatSper channel in heterologous systems. In rodents, sialyltransferase displays maturation-associated quantitative changes (*Ram et al., 1989*; *Scully and Shur, 1988*) and sperm lose sialic acid from the surface during capacitation (*Ma et al., 2012*). Sperm glycoproteins promote sperm migration and survival in the female reproductive tract (*Ma et al., 2016*). We speculate that mature CatSper1 in sperm contains terminal sialic acid residues, consistent with the small drop in Catsper1 molecular weight during capacitation. The dynamic sugar modifications on CatSper1 may serve as a binding site for decapacitation factors and/or a recognition site during capacitation. Supporting this idea, it was previously shown that mouse sperm lacking the CatSper channel cannot pass through the UTJ (*Chung et al., 2014*; *Ho et al., 2009*).

Capacitation-associated CatSper1 degradation is blocked by incubation with a 26S proteasome inhibitor, MG-132 (*Chung et al., 2014*). Now we show that solubilized sperm membrane fraction contains additional proteolytic activities that cleave within CatSper1 NTD. The proteolysis involves two distinct pathways: $Ca^{2+}$ entry and phosphorylation cascades. We hypothesize that a member of calpains, the $Ca^{2+}$ dependent modulatory protease family, might cleave CatSper1, as their proteolytic activity can be regulated by PKA (*Du et al., 2018*). Among 15 calpain proteins identified in mammals (*Ono et al., 2016*), calpain1 and calpain11 were previously detected in our sperm proteome (*Hwang et al., 2019*). Intriguingly, we observed that CatSper1 processing requires not a simple rise in intracellular $Ca^{2+}$, but rather $Ca^{2+}$ influx mediated by the CatSper channel which normally accompanies membrane reorganization during capacitation. Increased $Ca^{2+}$ level overrides the phosphorylation effect on calpain1 activity (*Du et al., 2018*). We speculate that calpain11 might be similarly regulated to calpain1, as their domain structures and catalytic residues are conserved

(*Ono et al., 2016*). Since recombinant CatSper1 is cleaved more specifically by sperm lysates, we propose that the testis-specific calpain11 (*Ben-Aharon et al., 2006*) may target CatSper1.

The effect of CatSper1 truncation on channel activity and sperm motility remains to be determined in future studies. CatSper1 truncation may be coordinated with molecular changes of other CatSper subunits. For example, the protein level of CatSper2, but not CatSper3 or 4, also decreases after capacitation when probed with the antibody recognizing its C-terminal domain (CTD) (*Figure 1*; *Chung et al., 2014*). Since the cytoplasmic modulatory subunits, CatSperζ and Efcab9, mainly interact with the channel pore (*Hwang et al., 2019*), specific processing of the intracellular domains of pore subunits could alter the interactions and subsequent channel activity. Spermatozoa that successfully navigate to the fertilization site in the female reproductive tract and interact with the egg are recognized by intact CatSper1. CatSper1 processing may lead to a loss of control in hyperactivation and eventually end sperm life.

## Physiological function of capacitation-associated tyrosine phosphorylation and acrosome reaction

An increase in pY is one of the various capacitation-associated parameters observed from in vitro capacitated sperm cells (*Visconti et al., 1995*). Subsequently, pY was observed in the flagellum of mouse and human sperm interacting with the oocyte in the medium that supports sperm capacitation and fertilization in vitro (*Sakkas et al., 2003*; *Urner et al., 2001*). This correlation of pY and the zona binding previously established pY as an indicator of successful sperm capacitation. More recently, however, different observations have been made with in vivo approaches. In sows inseminated close to ovulation, spermatozoa found in the UTJ exhibited more phosphorylation in the flagella than those bound to oviductal epithelial cells (OEC), where pY was limited to the equatorial region in the sperm head or no pY was observed (*Luño et al., 2013*). In mice, the testis-specific tyrosine kinase, Fer, is demonstrated as a master kinase for capacitation-associated pY (*Alvau et al., 2016*). Surprisingly, homozygous *Fer*-mutant male mice are fertile even though their sperm do not develop pY. All together, these results lead to a new interpretation of the physiological significance of pY: successful sperm capacitation does not require pY development. Determining the precise time and place of pY development in sperm in situ would help to elucidate its function in sperm capacitation and fertilization. Here we have shown that sperm cells, which have capacitated in vivo and successfully migrated to the ampulla, are characterized, not only by intact CatSper1, but also by relative lack of pY development and reacted acrosome. These results coincide with our observations from in vitro capacitated sperm cells and other previous studies; pY development inversely correlates with CatSper1 integrity at the single cell level (*Figure 2*); genetic and pharmacological ablation of Ca$^{2+}$ entry potentiates pY (*Chung et al., 2014*; *Navarrete et al., 2015*); AR occurs in mid-isthmus before contacting an oocyte ZP( *Hino et al., 2016*; *Jin et al., 2011*; *Muro et al., 2016*).

Sperm remaining in the female reproductive tract need to be eliminated after fertilization. They may undergo apoptosis and phagocytosis in the female reproductive tract (*Aitken and Baker, 2013*; *Chakraborty and Nelson, 1975*) and/or become lost in the peritoneal cavity (*Mortimer and Templeton, 1982*). pY is reported to mediate apoptosis in immune cells (*Yousefi et al., 1994*) and cancer cells (*Liu et al., 1994*). We propose that capacitation-associated global pY development represents degenerating sperm which might concomitantly lose motility. It is intriguing that capacitation-associated reactive oxygen species (ROS) generation activates intrinsic apoptotic cascade and compromises sperm motility (*Koppers et al., 2011*). Consistent with this idea, ROS inactivates protein tyrosine phosphatase (*Tonks, 2005*) and enhances pY development in sperm (*Aitken et al., 1998*). Inhibition of PKA anchoring to AKAPs, which induces CatSper1 truncation and degradation, also suppresses acrosome reaction in capacitating sperm cells in vitro (*Stival et al., 2018*). Thus, CatSper-mediated Ca$^{2+}$ signaling directly or indirectly contributes to sperm acrosome reaction in the female tract. Future work will determine molecular mechanisms by which CatSper channel activity fine-tunes Ca$^{2+}$ signaling to regulate hyperactivated motility, as well as how the Ca$^{2+}$ signaling is linked to coordinate acrosome reaction.

## New in situ molecular imaging platform for the study of fertilization and reproduction

The successful development of in vitro capacitation and fertilization systems provided fundamental insights into sperm capacitation, fertilization, and early embryogenesis. On the other hand, it is evident that the in vitro systems have limitations. Sperm numbers required for IVF are much higher than those observed at the fertilization site in vivo (*Suarez, 2007*). Sperm capacitated in vitro do not encounter the anatomically and spatially distinct environment of the female reproductive tract, for example, missing their interaction with the oviductal epithelial cells. In vitro capacitation also lacks secretory factors from the male and female reproductive tracts that can affect the surface protein dynamics during the capacitation process (*Flesch and Gadella, 2000*). Mouse models that typically use epidydimal sperm for in vitro studies do not contain secretions from male glands. This is in contrast with ejaculated sperm from human and domestic animals. Recent studies have observed sperm behavior in the physiological context through ex vivo imaging of sperm in the mouse and bovine oviducts under transillumination (*Hino and Yanagimachi, 2019*; *Ishikawa et al., 2016*; *Kölle et al., 2009*; *Muro et al., 2016*). Yet this technique is limited in providing molecular information at a single cell level, as live imaging is not easily amenable to direct molecular labeling and 3D volume imaging.

Here, we report new systems to molecularly examine individual sperm cells capacitated in vivo. Polymerization of the hydrogel-embedded time-mated female reproductive tract followed by passive clearing provides a stable meshwork to minimally disturb the original location of sperm cells inside the female tract. This approach allowed us to assess the fine organization of CatSper nanodomains in the sperm cells distributed along the female reproductive tract. We showed that both the intensity and the quadrilateral detection of the domains probed by CatSper1 appear as the common pattern of sperm reaching the ampulla and potentially fertilizing the oocyte. The experimental outputs complement the molecular and functional information of sperm released from micro-dissected female tracts and ex vivo imaging, identifying molecular and functional signatures of fertilizing sperm in the physiological context. Furthermore, we demonstrate the efficacy of topological heat-map representations of cumulative results by automatic sperm detection and image post-processing and averaging; this method provides statistically robust presentation and interpretation of the volumetric image data.

The present study opens up new horizons to microscopically visualize and analyze molecular events in single sperm cells that achieve fertilization. This will allow us to better understand physiologically relevant cellular signaling pathways directly involved in fertilization. We also have illustrated that the same approach of tissue-clearing based 3D in situ molecular imaging is applicable to study gametogenesis in situ. Future areas for investigations as natural extensions of the current study are gameto-maternal interaction, development, transport, and implantation of early embryos and maternal-fetal communication. Developing gamete-specific antibodies and/or knockout validated antibodies to probe molecular abundancy and dynamics in situ and post-processing tools for various parameters will be critical to this end.

## Materials and methods

### Key resources table

| Reagent type (species) or resource | Designation | Source or reference | Identifiers | Additional information |
|---|---|---|---|---|
| Gene (*Mus musculus*) | *Catsper1* | GenBank | Gene ID: 225865 | |
| Gene (*Mus musculus*) | *Catsper1* | GenBank | Gene ID: 225865 | |
| Strain, strain background (*Escherichia coli*) | 10-b | New England BioLabs | Cat# C3019H | |
| Strain, strain background *Mus musculus* | B6.129S4-*Catsper1*tm1Clph/J | Laboratory of David E. Clapham (*Ren et al., 2001*) | RRID:IMSR_JAX:0 18311 | |

*Continued on next page*

*Continued*

| Reagent type (species) or resource | Designation | Source or reference | Identifiers | Additional information |
|---|---|---|---|---|
| Strain, strain background *Mus musculus* | B6D2-Tg(CAG/Su9 -DsRed2,Acr3-EGFP) RBGS002Osb/OsbRbrc | *Hasuwa et al., 2010* | RRID:IMSR_RBRC03743 | |
| Strain, strain background *Mus musculus* | C57BL/6 | Charles River Laboratories | Crl:029 | |
| Strain, strain background *Mus musculus* | B6D2F1 | Charles River Laboratories | Crl:099 | |
| Strain, strain background *Mus musculus* | CD1 | Charles River Laboratories | Crl:CD1(ICR) RRID:IMSR_CRL:022 | |
| Strain, strain background (*Escherichia coli*) | b-10 | New England BioLabs | CMC0016 | |
| Cell line (*Homo sapiens*) | 293T | ATCC | Cat# CRL-3216; RRID:CVCL_0063 | |
| Biological sample (*Mus musculus*) | Testes | Chung lab, Yale School of Medicine | N/A | |
| Biological sample (*Mus musculus*) | Spermatozoa | Chung lab, Yale School of Medicine | N/A | |
| Antibody | Rabbit polyclonal anti-mCatSper1 | Laboratory of David E. Clapham (*Ren et al., 2001*) | RRID:AB_2314097 | Western blot, (2 µg/mL); Immunocytochemistry, (10 µg/mL); In situ imaging, (7 µg/mL) |
| Antibody | Rabbit polyclonal anti-mCatSper3 | Laboratory of David E. Clapham (*Qi et al., 2007*) | N/A | (2 µg/mL) |
| Antibody | Rabbit polyclonal anti-mCatSperε | Laboratory of Jean-Ju Chung (*Chung et al., 2017*) | N/A | (1.6 µg/mL) |
| Antibody | Rabbit Polyclonal CA-IV antibody | SantaCruz | Cat# sc-25598 | (1:500) |
| Antibody | Mouse monoclonal anti-caveolin1 (clone 2297) | BD Biosciences | Cat# 610406 RRID:AB_397788 | (1:500) |
| Antibody | Mouse monoclonal anti-phosphotyrosine (clone 4G10) | EMD milipore | Cat# 05–321; RRID:AB_309678 | Western blot, (1:1,000); Immunocytochemistry, (1:1000) |
| Antibody | Mouse monoclonal anti-acetylated tubulin (clone 6-11b-1) | EMD milipore | Cat# 7451; RRID:AB_609894 | Western blot, (1:20,000); In situ imaging, (1:100) |
| Antibody | Rabbit monoclonal anti-beta actin (clone 13E5) | Cell signaling technology | Cat# 4970 RRID:AB_2223172 | (1:100) |
| Antibody | Rabbit monoclonal anti-HA (clone C29F4) | Cell signaling technology | Cat# 3724 RRID:AB_1549585 | (1:2,000) |
| Antibody | Goat polyclonal anti-mouse IgG-HRP | Jackson ImmunoResearch | Cat# 115-035-003; RRID:AB_10015289 | (1:10,000) |
| Antibody | Goat polyclonal anti-rabbit IgG-HRP | Jackson ImmunoResearch | Cat# 111-035-144; RRID:AB_2307391 | (1:10,000) |
| Antibody | Goat polyclonal anti-rabbit IgG-Alexa 568 | Invitrogen | Cat# A-11036 RRID:AB_10563566 | Immunocytochemistry, (1:1000); In situ imaging, (1:500) |
| Antibody | Goat polyclonal anti-rabbit IgG-Alexa 647 | Invitrogen | Cat# A-21245 RRID:AB_2535813 | Immunocytochemistry, (1:1000); In situ imaging, (1:500) |

*Continued on next page*

Continued

| Reagent type (species) or resource | Designation | Source or reference | Identifiers | Additional information |
|---|---|---|---|---|
| Antibody | Goat polyclonal anti-mouse IgG-Alexa 488 | Invitrogen | Cat# A-11029 RRID:AB_138404 | Immunocytochemistry, (1:1000); In situ imaging, (1:500) |
| Antibody | Goat polyclonal anti-mouse IgG-Alexa 647 | Invitrogen | Cat# A21236 RRID:AB_2535805 | Immunocytochemistry, (1:1000); In situ imaging, (1:500) |
| Antibody | Mouse monoclonal anti-HA agarose (clone HA-7) | EMD milipore | Cat# A2095 AB_257974 | (10 uL) |
| Antibody | Anti-HA magnetic beads (clone 2–2.2.14) | Thermo scietific | Cat# 88836 RRID:AB_2749815 | (5 uL) |
| Recombinant DNA reagent | pcDNA3-mCatsper1 | Laboratory of David E. Clapham | N/A | |
| Recombinant DNA reagent | pcDNA3.1(-)-Flag-Catsper1-FL-HA | This paper | N/A | Encoding Mouse CatSper1, 1–686 aa |
| Recombinant DNA reagent | pcDNA3.1(-)-Flag-Catsper1-ND-HA | This paper | N/A | Encoding Mouse CatSper1, 345–686 aa |
| Commercial assay or kit | O-glycosidase | New England BioLabs | Cat# P0733S | (80,000 Unit) |
| Commercial assay or kit | PNGase F | Sigma-Aldrich | Cat# P9120 | (2 nM Unit) |
| Peptide, recombinant protein | Protein phosphatase 1 (PP1) | New England BioLabs | Cat# P0754 | (0.1 Unit) |
| Peptide, recombinant protein | Protein tyrosine phosphatase 1B (PTP1B) | Abcam | Cat# Ab42574 | (5 Unit) |
| Chemical compound, drug | Sodium orthovanadate | New England BioLabs | Cat# P0758S | (1 mM) |
| Chemical compound, drug | H89 | Calbiochem | Cat# 371963 | (50 µM) |
| Chemical compound, drug | ST-Ht31 | Promega | Cat# V8211 | (10 µM) |
| Chemical compound, drug | Calyculin A | Calbiochem | Cat# 208851 | (100 nM) |
| Chemical compound, drug | A23187 | Sigma-Aldrich | Cat# C7522 | (10 µM) |
| Chemical compound, drug | Calpain inhibitor I | Calbiochem | Cat# 208719 | (20 µM) |
| Chemical compound, drug | Calpain inhibitor II | Enzo life science | Cat# BML-PI100 | (20 µM) |
| Chemical compound, drug | Calpain inhibitor III | Enzo life science | Cat# BML-PI130 | (20 µM) |
| Chemical compound, drug | PNA-Alexa 568 | Invitrogen | Cat# L32458 | (1:500) |
| Chemical compound, drug | WGA-Alexa 555 | Invitrogen | Cat# W32464 | (1:500) |
| Chemical compound, drug | WGA-Alexa 647 | Invitrogen | Cat# W32466 | (1:500) |
| Chemical compound, drug | Azo-initiator | Wako | Cat# VA-044 | |
| Chemical compound, drug | 32% Paraformaldehyde | Electron Microscopy Sciences | Cat# 15714 s | |
| Chemical compound, drug | 50% Glutaraldehyde | Electron Microscopy Sciences | Cat# 16310 | |

*Continued*

| Reagent type (species) or resource | Designation | Source or reference | Identifiers | Additional information |
|---|---|---|---|---|
| Chemical compound, drug | Electrophoretic Tissue Clearing Solution | Logos Biosystems | Cat# C13001 | |
| Chemical compound, drug | N-methyl-d-glucamine | Sigma-Aldrich | Cat# M2004 | |
| Chemical compound, drug | Diatrizoic acid | Sigma-Aldrich | Cat# D9268 | |
| Chemical compound, drug | 60% Iodixanol | Sigma-Aldrich | Cat# D1556 | |
| Chemical compound, drug | DAPI | Thermo scietific | Cat# D1306 | 1:1000 |
| Chemical compound, drug | Hoechst | Thermo scietific | Cat# 62249 | 1 µg/mL |
| Chemical compound, drug | Acrylamid | BioRad | Cat# 1610140 | |
| Software, algorithm | ImageJ | National Institue of Health | Public domain: https://imagej.nih.gov/ij/ | |
| Software, algorithm | Zen software | Carl Zeiss AG | ZEISS ZEN lite www.zeiss.com | |
| Software, algorithm | Imaris | BitPlane; Oxford instruments | Imaris 9.6 | |
| Software, algorithm | MATLAB | MathWorks | MATLAB 9.3 (R2017b) | |
| Software, algorithm | Blender | Blender Foundation, community | Blender 2.79 www.blender.org | |
| Software, algorithm | GraphPad | GraphPad Software | GraphPad Prism 5 | |
| Software, algorithm | ANNs development source materials | This paper | Source archive attached to *Figure 5* | Set of ANN development tools |
| Software, algorithm | CatSper quadrilateral structure 3D in situ analytical tools | This paper | Source archive attached to *Figure 6* | Set of 3D CatSper analytical tools |

## Animals

*Catsper1*-null (*Ren et al., 2001*) and Su9-DsRed2/Acr-EGFP (*Hasuwa et al., 2010*) mice were generated in the previous study and maintained on a C57BL/6 background. Su9-DsRed2/Acr-EGFP mice were crossbred with *Catsper1*-null mice to generate Su9-DsRed2/Acr-EGFP *Catsper1*-null mice. WT C57BL/6 and B6D2F1 male and CD1 female mice were purchased from Charles River Laboratories (Wilmington, MA). Mice were cared for in compliance with the guidelines approved by the Yale Animal Care and Use Committees.

## Mammalian cell lines

HEK293T and COS-7 cells were purchased from ATCC and authenticated by positive detection of SV40 DNA and STR-based analysis using the Cell ID System (Promega). They were cultured in DMEM (GIBCO) supplemented with 10% FBS (Thermofisher) and 1× Pen/Strep (GIBCO) at 37°C, 5% $CO_2$ condition. Cultured cells were used to express recombinant proteins (HEK293T cells) or make total cell lysates (COS-7 cells).

## Antibodies and reagents

In-house rabbit polyclonal CatSper1 (*Ren et al., 2001*), CatSper3 (*Qi et al., 2007*), CatSperε (*Chung et al., 2017*) antibodies were described previously. Polyclonal CA-IV antibody (M-50) was purchased from Santacruz. Monoclonal antibodies were purchased from BD Biosciences: anti-caveolin1 (clone 2297); EMD Milipore: anti-phosphotyrosine (clon4G10), anti-acetylated tubulin (clone 6-11B-1), anti-HA agarose (clone HA-7); Thermo Scientific: anti-HA magnetic beads; and Cell Signaling

Technology: β-actin (clone 13E5) and HA (clone C29F4). HRP-conjugated goat anti-rabbit IgG and goat anti-mouse IgG were from Jackson Immunoresearch. PNA-Alexa 568, WGA-Alexa 555, WGA-Alexa 647, goat anti-mouse IgG (Alexa 488 or 647), and goat anti-rabbit IgG (Alexa 568 or Alexa 647) were from Invitrogen. H89, calyculin A, and calpain inhibitor I were purchased from Calbiochem. ST-Ht31 was from Promega. Calpain inhibitor II and III were from Enzo life science. All other chemicals were from Sigma-Aldrich unless indicated.

## Epididymal sperm collection and in vitro capacitation

Sperm cells were released from caput, corpus, or cauda regions of the epididymis in M2 medium (EMD Millipore). To induce capacitation, sperm from caudal epididymis were incubated in human tubular fluid (HTF) medium or M16 (EMD Milipore) containing 25 mM sodium bicarbonate at 37°C, 5% $CO_2$ condition at $2 \times 10^6$ cells/mL concentration for the indicated time. Sperm cells were incubated under capacitating conditions with or without the following chemicals: H89 (50 μM), ST-Ht31 (10 μM), Calyculin A (100 nM), calpain inhibitor I (20 μM), calpain inhibitor II (20 μM), or calpain inhibitor III (20 μM). Sperm cells suspended in M2 medium ($2 \times 10^6$ cells/mL) were incubated with A23187 (10 μM) to induce $Ca^{2+}$ influx under non-capacitating conditions.

## Molecular cloning

NEB10β bacterial strain (NEB) was used for molecular cloning. Genomic regions encoding full-length (FL, 1–686 aa) and N-terminal domain deleted (ND, 345–686 aa) mouse CatSper1 were amplified from mouse CatSper1 expression vector (Hwang et al., 2019). The PCR products were subcloned into pcDNA3.1(-) vector using NEBuilder HiFi DNA Assembly (NEB) to express the recombinant proteins tagged with HA at C-terminus (pcDNA3.1(-)-FL-Catsper1-HA and pcDNA3.1(-)-ND-Catsper1-HA).

## Recombinant protein expression

HEK293T cells were transfected with constructs encoding FL-CatSper1 or ND-CatSper1 to express the recombinant proteins transiently. Polyethyleneimine was used for the transfection following the manufacturer's instruction as previously.

## Protein preparation and western blot

### Total protein extraction

Total proteins were extracted from sperm, testis, and cultured mammalian cells as previously described (Hwang et al., 2019). In short, collected epididymal sperm cells were washed with PBS and lysed in 2× LDS sampling buffer for 10 min at room temperature with agitation (RT). The whole sperm lysates were centrifuged at $14,000 \times g$ for 10 min at 4°C. Testes were homogenized in 0.32M sucrose and centrifuged at $1000 \times g$ for 10 min at 4°C to remove cell debris and nuclei. 1% Triton X-100 in PBS containing protease inhibitor cocktail (cOmplete, EDTA-free, Roche) was added to the cleared homogenates to make total testis lysate. The lysates were centrifuged at 4°C, $14,000 \times g$ for 30 min and the supernatant was used for the downstream experiments. Transfected HEK29T cells and COS-7 cells were washed and lysed with 1% Triton X-100 in PBS with protease inhibitor cocktail (Roche) at 4°C for 1 hr. Cell lysates were centrifuged at $14,000 \times g$ for 30 min. All the solubilized protein lysates from the sources described above were reduced by adding dithiothreitol (DTT) to 50 mM and denature by heating at 75°C for 5 min (testis and cultured cells) or 10 min (sperm).

### Discontinuous sucrose density gradient centrifugation

Discontinuous sucrose density gradient centrifugation was performed as previously described (Kaneto et al., 2008). To isolate and solubilize membrane fraction without using a detergent, cauda epididymal sperm cells washed and suspended in PBS ($1.0 \times 10^8$ cells/mL) were sonicated 3 times for 1 s each. Sonicated sperm cells were then centrifuged at $5000 \times g$ for 10 min at 4°C and the solubilized membrane fraction in the supernatant was collected. The solubilized membrane fraction was pelleted by ultracentrifugation at $100,000 \times g$ for 1 hr at 4°C and resuspended with PBS. The membrane suspension was mixed with an equal volume of 80% sucrose in PBS. A discontinuous sucrose gradient was layered with the 40%, 30%, and 5% sucrose solution from bottom to top in a tube discontinuously. The gradient was ultracentrifuged at $200,000 \times g$ for 20 hr at 2°C. Proteins collected

from each fraction were precipitated with 5% of trichloroacetic acid, ethanol washed, and dissolved in SDS sampling buffer.

## Dephosphorylation of sperm membrane proteins

Sperm membrane fractions from $1 \times 10^6$ sperm cells prepared as above were treated with protein phosphatase 1, (PP1, 0.1 unit; NEB), protein tyrosine phosphatase (PTP1B, 5 units; Abcam), or sodium orthovanadate ($Na_3VO_4$, 1 mM; NEB) to test dephosphorylation of CatSper1. The membrane fractions were incubated with the phosphatases or $Na_3VO_4$ in a reaction buffer containing 20 mM HEPES, 0.1 mM EDTA, and 0.1 mM DTT at 30℃ for the indicated times. The isolated sperm membrane was solubilized by adding Triton X-100 to the final 0.1% in PBS (PBS-T) for the indicated times at RT.

## Enzymatic deglycosylation

Glycosylation of CatSper1 from cauda sperm was tested using PNGase F (Sigma-Aldrich) and O-glycosidase (NEB) according to the manufacturer's instructions. Sperm cells ($4 \times 10^6$ cells) were washed with $1\times$ reaction buffer for each enzyme by centrifugation at $800 \times$ g for 3 min. Sperm pellets were re-suspended with each $1\times$ reaction buffer (20 mM or 50 mM sodium phosphate, pH 7.5 for PNGase F and O-glycosidase, respectively) and followed by sonication and centrifugation to collect sperm membrane fraction as described above. Collected supernatants were incubated with denaturation buffer at 100℃ for 5 min to denature glycoproteins before subject to enzymatic deglycosylation. The denatured sperm membrane fractions were incubated with detergent buffer (0.75% IGEPAL CA for PNGase F; 1% NP-40 for O-glycosidase) and each glycosidase (PNGase F, 2 nM unit; O-glycosidase, 80,000 unit) at 37℃ for 1 hr. All the enzyme-treated samples were mixed with LDS sampling buffer and denatured after adding DTT to 50 mM at 75℃ for 2 min.

## In Vitro proteolysis with sperm lysate

Proteolysis of the recombinant CatSper1 protein by sperm lysate was performed as previously described (Chung et al., 2014). Solubilized recombinant FL-CatSper1 and ND-CatSper1 were pulled-down with anti-HA agarose (EMD Millipore) for 1 hr at RT. The enriched recombinant proteins were incubated with 30 μL of sperm lysates solubilized from $3.0 \times 10^5$ sperm cells at 37℃ for the indicated times. Sperm lysates were prepared by sonication and incubation in PBS-T without protease inhibitor at 4℃ for 1 hr. After incubation, the mixture of recombinant protein and sperm lysates were mixed to $2\times$ LDS sampling buffer and denatured by adding DTT to 50 mM at 75℃ for 10 min.

## Western blot

Denatured protein samples were subjected to SDS-PAGE. Rabbit polyclonal CatSper1 (2 μg/mL), CatSper3 (2 μg/mL), CatSperε (1.6 μg/mL), and CAIV (1:500) antibodies and monoclonal HA (clone C29F4; 1:2,000), caveolin1 (clone 2297, 1:500), acetylated tubulin (clone 6-11B-1; 1:20,000), and phosphotyrosine (clone 4G10; 1:1000) antibodies were used for western blot. Anti-mouse IgG-HRP (1:10,000) and anti-rabbit IgG-HRP (1:10,000) were used for secondary antibodies.

## Sperm migration assay

Sperm migration assay was performed as previously described (Chung et al., 2014). Briefly, female mice were introduced to single-caged Su9-DsRed2/Acr-EGFP males for 30 min and checked for the vaginal plug. Whole female reproductive tracts were collected 8 hr post-coitus and subjected to ex vivo imaging to examine spermatozoa expressing reporter genes in the tract (Eclipse TE2000-U, Nikon).

## Collection of in vivo capacitated sperm

Female reproductive tracts from timed-mated females to Su9-DsRed2/Acr-EGFP or Su9-DsRed2/Acr-EGFP *Catsper1*-null males were collected 8 hr post-coitus. Sperm cells were released by micro-dissection of the female reproductive tract followed by lumen flushing of each tubal segment (cut into ~1–2 mm pieces). Each piece was placed in 50 μL of PBS on glass coverslips and the intraluminal materials were fixed immediately by air-dry followed by 4% PFA in PBS. Ampulla and uterine tissue

close to UTJ were placed in 100 µL of PBS and vortexed briefly to release the sperm within the tissues. Fixed sperm cells were subjected to immunostaining.

## Sperm immunocytochemistry

Non-capacitated or in vitro capacitated sperm cells on glass coverslips were washed with PBS and fixed with 4% paraformaldehyde (PFA) in PBS at RT for 10 min. Fixed samples were permeabilization with PBS-T for 10 min and blocked with 10% normal goat serum in PBS for 1 hr at RT. Blocked sperm cells were stained with primary antibodies, anti-CatSper1 (10 µg/mL), and anti-phosphotyrosine (1:1000), at 4°C for overnight, followed by staining with secondary antibodies for 1 hr at RT. Hoechst was used for counterstaining sperm head. Sperm cells were mounted (Vectashield, Vector Laboratories) and imaged with confocal microscopes (Zeiss LSM710 Elyra P1 and Olympus Fluoview 1000).

## Tissue clearing and molecular labeling of the cleared tissues

### CLARITY method

All 3D volume images from the main figures (*Figures 3* and *4*) were taken from female tracts subjected to the CLARITY method (*Chung et al., 2013*) with slight modification by clearing tissue-hydrogel passively without involving electrophoresis. Timed-mated females (8 hr post-coitus) and males were subjected to transcardiac perfusion using a peristaltic pump. The mice were perfused with each 20 mL of ice-cold PBS followed by freshly prepared hydrogel monomer solution (4% acrylamide, 2% Bis-acrylamide, 0.25% Azo-inhibitor [VA-044, Wako], 4% PFA in PBS). The whole female tract or testis-hydrogel were dissected from animals after perfusion and placed in 10 mL of fresh hydrogel monomer solution for post-fixation. The collected tissues in monomer solution were heated at 37°C with degassing for 15 min, followed by incubation at 37°C for 2–3 hr for tissue gelation. The gelated tissues were washed with a clearing solution containing 200 mM boric acid and 4% sodium dodecyl sulfate (pH 8.5) three times for 24 hr each by gentle rocking at 55°C. Cleared tissues were further washed with PBS-T for 24 hr. The cleared female tracts were subjected to dye- and/or immunolabeling: the cleared tissues were incubated with CatSper1 (7 µg/mL) or AcTub (1:100) antibodies in PBS-T overnight at RT, followed by washing with PBS-T for 24 hr. Washed samples were stained with the secondary antibodies (1:500) overnight. Fluorescence dye conjugated PNA or WGA was used to detect sugar residues (1:1000) and DAPI was used for counterstaining (1:1000) in PBS-T. Stained tissues were washed and refractive index-matched in RIMS solution (*Chung et al., 2013*) overnight. The index-matched samples were placed on an imaging chamber filled with RIMS solution and imaged. All cleared tissues were imaged with a laser scanning microscope (Zeiss LSM710 Elyra P1). EC plan-Neofluar 10×/0.3, LD LCI Plan-Apochromat 40 × 1.2, and Plan-Apochromat 63×/1.4 objectives were used for imaging. Tile scanning and z-stacking for volume imaging were carried out with functions incorporated in Zen black 2012 SP2 (Carl Zeiss) and Zen blue 2011 SP1 software (Carl Zeiss) was used for 3D rendering.

### X-CLARITY

Ovary, testis, and epididymis images (*Figure 3—figure supplement 1A,D–I*) were taken using the X-CLARITY method, following manufacturer's instructions (Logos biosystems). Animals transcardially fixed with 4% PFA were post-fixed in the fresh fixative for 4–6 hr. The post-fixed tissues were then immersed in a modified hydrogel solution (4% acrylamide, 0.25% Azo-inhibitor [VA-044, Wako], 4% PFA in PBS) for 4–6 hr. The samples were degassed and polymerized as described in the CLARITY method. The gelated tissues were washed with PBS and placed in an electrophoretic tissue clearing (ETC) chamber. Tissues in the ETC chamber were cleared by clearing solution described above with active electrophoretic forcing of tissue for 6–8 hr. Cleared tissues were washed and stained with β-actin and WGA.

### PACT-PRESTO

3D volume images of the oviduct and UTJ (*Figure 3—figure supplement 1B,C*) were obtained from the female tract cleared by modified passive ACT-PRESTO (*Lee et al., 2016*) without involving electrophoretic clearing. In brief, female tracts were fixed in 4% PFA by transcardiac perfusion, followed by post-fixation in fresh fixative solution for 4–6 hr at 4°C. The post-fixed samples were incubated in the modified hydrogel monomer solution without additional fixative (4% acrylamide, 0.25% Azo-

inhibitor in PBS) 4–6 hr at 4°C. The samples were degassed and polymerized at 37°C as described in the CLARITY method. Hydrogel-infused tissues were cleared with the clearing solution. The cleared tissues were washed with PBS overnight at RT and facilitated labeling is achieved by vacuum-applied negative pressure.

### SWITCH method

3D volume images of the testis (*Figure 3—figure supplement 1F*) were taken from male mice trans-cardially perfused and cleared by the SWITCH method (*Murray et al., 2015*). Fixed testes by SWITCH fixative (4% PFA, 1% glutaraldehyde (GA) in PBS) were washed with PBS-T and quenched by 4% glycine and 4% acetamide in PBS at 37°C overnight. The quenched samples were passively cleared with SWITCH solution (200 mM SDS, 20 mM $Na_2SO_3$, 10 mM NaOH, pH 9.0) two times at 60°C for 3 hr each. The cleared tissues were washed with PBS-T for 12–24 hr at 37°C and incubated with refraction index-matching solution (RIMS: 29.4% diatrizoic acid, 23.5% n-methyl-d-glucamine, 32.4% iodixanol). The index-matched samples were mounted and imaged.

## Artificial neural network (ANN) image processing

The overall strategy for the artificial neural network (ANN) image processing is described in *Figure 5—figure supplement 2A*. The individual signal patterns (sperm, somatic cell nuclei, and noise) were isolated from the original volume images using Zen Blue (Carl Zeiss) and IMARIS software (Oxford instruments) and exported as. obj/.fbx files. The isolated signal patterns were used to generate 3D training environments for ANN by importing different abundancies of the individual components (*Figure 5—figure supplement 1*) to the 3D environment operating system, Blender 2.79 (https://www.blender.org/); the individual 3D training environment ($\sim10^4$) generated together with the exactly defined coordinates of individual components were exported as. obj/.fbx/Notepad++ files. The ANN training environments were used to develop the ANN detecting the sperm in situ. The ANN training was carried out using MATLAB 9.3 (R2017b) software ANN toolbox. The input to the ANN would be virtual *z*-stacks of the produced training environments. The isolated sperm signal patterns were used as a target signature. The supervised training process was performed by comparing the vector coordinates of the individual sperm signatures in the output with the pre-defined vector coordinates of the signatures in the input. This approach also enabled us to evaluate the ANN performance and to quantify signature detection sensitivity and specificity (*Figure 5—figure supplement 2A*), panel 6 'ANN performance evaluation'. The detection sensitivity and specificity of ANNs were the major performance indicators used to select ANNs. ANNs with the best performance in detecting the sperm signature were subsequently applied to detect the sperm fluorescent signatures and their post-processing in real volumetric data. In the real environments, selected ANNs showed both sensitivity and specificity around 90% (*Figure 5—figure supplement 2C*). See the related source data.

## Statistical analyses

Statistical analyses were carried out with a one-way analysis of variance (ANOVA) with the Tukey post hoc test. Differences were considered significant at $p < 0.05$. For ANN analysis, both parametric (ANOVA; Tukey post hoc) and non-parametric (KW-ANOVA) tests were carried out to evaluate the presented differences; both tests resulted in the same significance output with differences considered significant at $p < 0.05$.

## Acknowledgements

We thank David E Clapham and Katerina Komrskova for sharing resources, Kwanghun Chung for discussions on tissue clearing methods, Luke L McGoldrick for critical reading of the draft, Logos Biosystems for use of X-CLARITY system, and MFF computational core facility for assistance in ANN analysis. This work was supported by start-up funds from Yale University School of Medicine, a Yale Goodman-Gilman Scholar Award-2015, and NIH (R01HD096745) to J-JC; by the Czech Science Foundation (GACR No. GJ20-17403Y) to LD; by project BIOCEV (CZ.1.05/1.1.00/02/0109) from the ERDF and the Institute of BiotechnologyRVO:86652036. JYH is a Male Contraceptive Initiative post-doctoral fellow.

## Additional information

### Funding

| Funder | Grant reference number | Author |
|---|---|---|
| National Institutes of Health | R01HD096745 | Jean-Ju Chung |
| Yale School of Medicine | Start-up funds | Jean-Ju Chung |
| Yale University | Yale Goodman-Gilman ScholarAward-2015 | Jean-Ju Chung |
| Male Contraceptive Initiative | Postdoctoral fellowship | Jae Yeon Hwang |
| Czech Science Foundation | GJ20-17403Y | Lukas Ded |

The funders had no role in study design, data collection and interpretation, or the decision to submit the work for publication.

### Author contributions
Lukas Ded, Data curation, Software, Formal analysis, Validation, Visualization, Methodology, Writing - original draft, Project administration, Performed microdissection, 3D in situ imaging, ANN-based analysis and rendering of in situ images, assembled figures and wrote the manuscript together with J-JC; Jae Yeon Hwang, Data curation, Formal analysis, Validation, Visualization, Methodology, Writing - original draft, Performed and analyzed biochemistry experiment, confocal imaging of in vitro experiments, assembled figures and wrote the manuscript together with J-JC; Kiyoshi Miki, Data curation, Validation, Methodology, Performed and analyzed biochemistry experiments; Huanan F Shi, Data curation, Visualization, Methodology, Contributed to 3D in situ imaging; Jean-Ju Chung, Conceptualization, Resources, Data curation, Formal analysis, Supervision, Funding acquisition, Validation, Investigation, Visualization, Methodology, Writing - original draft, Project administration, Writing - review and editing, Conceived and supervised the project, Designed, performed and analyzed experiments: various tissue clearing applications, confocal imaging of in vitro and ex vivo experiments, Assembled figures and wrote the manuscript with the input from the co-authors

### Author ORCIDs
Lukas Ded (iD) https://orcid.org/0000-0003-1053-4025
Jae Yeon Hwang (iD) https://orcid.org/0000-0002-6493-4182
Huanan F Shi (iD) http://orcid.org/0000-0003-3710-5917
Jean-Ju Chung (iD) https://orcid.org/0000-0001-8018-1355

### Ethics
Animal experimentation: Animal experimentation: This study was performed in strict accordance with the recommendations in the Guide for the Care and Use of Laboratory Animals of the National Institutes of Health. All the mice were treated in accordance with guidelines approved by Yale (20079) Animal Care and Use Committees (IACUC).

### Decision letter and Author response
Decision letter https://doi.org/10.7554/eLife.62043.sa1
Author response https://doi.org/10.7554/eLife.62043.sa2

## Additional files

### Supplementary files
• Transparent reporting form

## Data availability

All data generated or analysed during this study are included in the manuscript, supplementary and source data files.

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
