## [Decision Letter]

**Acceptance summary:**

This manuscript reports a novel in-situ molecular imaging technique for characterizing sperm physiology along the whole reproductive tract at a very high spatial and temporal resolution. Out of millions of sperm deposited in the reproductive tract upon coitus, very few reach the fertilization site, and only one succeeds in fertilizing the egg. The low success rate guarantees selection of high "quality" sperm and, in part, guards against polyspermy, but the characteristics of the "successful" spermatozoa have not been well defined. The findings reported here provide unprecedented insights into this longstanding question.

**Decision letter after peer review:**

Thank you for submitting your article "3D in situ imaging of female reproductive tract reveals molecular signatures of fertilizing spermatozoa in mice" for consideration by *eLife*. Your article has been reviewed by two peer reviewers, and the evaluation has been overseen by Merritt Maduke as the Reviewing Editor and Richard Aldrich as the Senior Editor. The reviewers have opted to remain anonymous.

The reviewers have discussed the reviews with one another and the Reviewing Editor has drafted this decision to help you prepare a revised submission.

Summary:

In this manuscript, Ded et al. report findings using elegant in-situ molecular imaging of spermatozoa along the whole female reproductive tract using micro dissection, cleared tissues and artificial neural network-based automatic detection. Out of millions of sperm deposited in the reproductive tract upon coitus, very few reach the fertilization site, and only one succeeds in fertilizing the egg. The low success rate guarantees selection of the high "quality" of sperm and, in part, guards against polyspermy, but the characteristics of the "successful" spermatozoa have not been well defined. The authors developed a novel technique of in-situ molecular imaging along the whole reproductive tract and provided some of the most convincing answers to the long-standing question.

Ded and colleagues explore how CatSper channels change between the initial formation of the sperm, after spending time in the female reproductive tract, and the sperm near the site of fertilization. CatSper conducts a pH regulated, Ca^2+^ current through sperm, and is required for a series of processes called capacitation. Ca^2+^ entry via CatSper is specifically required for a sperm swimming pattern called hyperactivation, and in human sperm, a CatSper-conducted Ca^2+^ entry is hypothesized to signal the acrosome reaction. The channel is a multimeric complex comprised of four different alpha subunits, and several accessory proteins that are nearly all required for CatSper expression and male fertility. Ded et al. found that most sperm lose the intact sperm Ca^2+^ channel CatSper in the reproductive tract, but the ones that reach the fertilization sites have intact channels. Surprisingly, the sperm at the site do not have tyrosine phosphorylation, which is acquired during capacitation and traditionally thought to be important for sperm physiology. Also surprising is the finding that sperm successfully reaching the site of fertilization have already undergone the acrosome reaction (AR), a process during which the released acrosomal enzymes have been thought to help digest the egg for the sperm to penetrate and fuse with the egg. Intriguingly, the authors also found that CatSper1 proteins in sperm and in the precursors have different molecular weights. The authors attribute the difference, at least partly, to Ca^2+^-dependent protease-mediated cleavage/degradation. Despite its fundamental importance in fertilization and clear demonstration to be a Ca^2+^ channel in its native environment (sperm), CatSper has failed to be successfully reconstituted in heterologous expression system as a functional ion channel. The finding of molecular modification in the CatSper protein in this paper may help future functional reconstitution.

Overall, this manuscript presents new methods and provides molecular characterization of sperm physiology along the whole reproductive tract at a very high spatial and temporal resolution. The novel findings are convincing and important for our understanding of mammalian fertilization and ion channel-mediated calcium signaling. Suggested revisions will improve and clarify the presentation.

Essential revisions:

1) This manuscript reports several findings that will be of interest to the broad scientific community. However, it is written for a sperm-focused audience, rather than the *eLife* readership. The authors should clarify writing to increase the accessibility of this manuscript by stating motivations for each experiment in the context of biological processes. Also, the manuscript would be easier to read if there was a cohesive, central hypothesis that connect the seemingly separate stories (glycosylation, independence of tyrosine phosphorylation from CatSper, and the geometry of sperm distribution in different parts of the female reproductive tract).

2) "The molecular weight and amount of CatSper1, but not the other CatSper subunits, declines during sperm capacitation (Chung et al., 2014; Figure 1B, D)."

It is not clear from the figure that there is a decline in molecular weight of CatSper1 during capacitation. Similarly, it is not clear there is a decrease in amount, as the amount in the figure is not normalized. If there were a specific decrease in amount of CatSper1, but not that of the other CatSpers, would it suggest that the stoichiometry of the CatSper channel changes during capacitation?

3) Figure 1B, the figure legend for this panel is incomplete. In its current form, it is hard for the readers to understand what are the treatments in the panel.

4) Figure 1B, in the treatment with O-Gly, the cleavage appears quite incomplete. Did the density of the lower band increase with longer treatment?

5) Figure 1G, the legend states that "ND-CatSper1 proteins remain largely unchanged under the same conditions (bottom)", but there seems to be significant change in the figure. Please show statistics from several blots.

6) Figure 1—figure supplement 1A: The figure legend suggests that CatSper1 is not a phosphoprotein, yet only tyrosine phosphorylation was explored. Does PP1 dephosphorylate serine/threonine as well? If not, the text should be updated to indicate that CatSper1 is not tyrosine phosphorylated.

7) Figure 1—figure supplement 1E: Please indicate how you determined that A23178 induced CatSper1 processing occurs via the same pathway as PKA. Perhaps the authors have demonstrated that H89 prevents A23178 induced CatSper1 degradation? Or inhibit PKA signaled degradation by loading sperm with BAPTA-AM? Alternatively, it is possible that these be two pathways are distinct? If this is not definitively known, the authors should clarify the schematic to indicate that these steps are speculated.

8) The motivation for use of the Su9DsRed; Acr EGFP mice is never stated in the manuscript. The authors should state why these sperm were used in both the text, and figure labels should be also be updated (e.g. Figure 2D – to which proteins/structures are these fluorophores anchored?).

9) Figure 2: This figure would be substantially enhanced by inclusion of a pie chart for in vivo capacitated sperm, similar to the chart included for in vitro capacitated sperm. Also, did you look at pY in the cleared organs?

10) When describing the images in Figure 3, the text (subsection “3D in situ molecular imaging of gametes in the female reproductive tract”, first paragraph) should state the protein/tissues labeled by each stain.

11) Have you quantified tyrosine phosphorylation of in vivo sperm? Perhaps the relative fluorescence of immune-stained sperm? The concluding statement to the in vivo imaging paragraph (subsection “Sperm cell that successfully reach the ampulla are CatSper1-intact and acrosome reacted”) suggests that you do. Perhaps the evidence could be explicitly stated and readers directed to a figure panel. Again, including a pie chart with this data would be of great value to the readers (in Figure 2E).

12) The methods describing development of ANN for these experiments is missing information such as the criteria used for choosing ANN? What quantitative evaluations were used in making these decisions? What percentage of the data were set aside for training?

13) Were the sperm in Figure 6 (used for ANN analysis) immunolabeled for CatSper1?

14) Figure 6H. Left panel, what do the delta values refer to? Fluorescence intensity? Right panel: delta fluorescence, what values were used for the basal measurements?

15) A role for calpain proteolysis in CatSper1 degradation with in vitro capacitation is intriguing, however, it is not clear from the writing that this is a hypothesis. If this is not a hypothesis and Calpain is known to mediate CatSper1 proteolysis, a citation should be provided. If not, the writing should be edited to clarify that this is a hypothesis.

---

## [Author Response]

Essential revisions:1) This manuscript reports several findings that will be of interest to the broad scientific community. However, it is written for a sperm-focused audience, rather than the eLife readership. The authors should clarify writing to increase the accessibility of this manuscript by stating motivations for each experiment in the context of biological processes. Also, the manuscript would be easier to read if there was a cohesive, central hypothesis that connect the seemingly separate stories (glycosylation, independence of tyrosine phosphorylation from CatSper, and the geometry of sperm distribution in different parts of the female reproductive tract).

Thank you for this constructive suggestion. We have clarified writing for the broad readership of *eLife*. We have stated our central hypothesis more clearly in Introduction (third paragraph) and revised the text to explicitly state our motivations for each experiment in the Results (subsections “CatSper1 undergoes post-translational modifications during sperm development and maturation”, “CatSper1 resides in the subdomains of lipid rafts in mature sperm and is processed during capacitation”, “CatSper1 degradation correlates with pY development in sperm cells capacitated in vitro” and “Sperm cells capacitated in vivo become heterogeneous along the female tract with distinct molecular characteristics”). These changes indeed improved the presentation of our findings and will increase the accessibility of the manuscript.

2) "The molecular weight and amount of CatSper1, but not the other CatSper subunits, declines during sperm capacitation (Chung et al., 2014; Figure 1B, D)."It is not clear from the figure that there is a decline in molecular weight of CatSper1 during capacitation. Similarly, it is not clear there is a decrease in amount, as the amount in the figure is not normalized. If there were a specific decrease in amount of CatSper1, but not that of the other CatSpers, would it suggest that the stoichiometry of the CatSper channel changes during capacitation?

Re: a decline in molecular weight: Sperm CatSper1 proteins are detected as two close bands, which are best resolved only when the samples are separated in a low percentage gel for an extended time. We speculate that the small difference in molecular weight comes from terminal sialylation on O-glycosylated CatSper1. To address the reviewer’s concern, we have re-arranged Figure 1B to show sperm lanes (WCL and Membrane) from a short-exposed image that clearly demonstrates that the upper band specifically declines in WCL from capacitated sperm. This small drop in the molecular weight of sperm CatSper1 is further supported by two other blots (Figure 1I and K) in which AcTub serves as a loading control. We have revised the corresponding figure legend accordingly.

Re: a decrease in amount and specificity: We previously reported normalized protein levels of CatSper subunits after capacitation to show specific decrease of CatSper1 (~60% of non-capacitated, ***p*<0.01) and CatSper2 to a lesser extent (Chung et al., 2014; Figure 6B and Supplementary Figure 7D). In this study, CA IV in Figure 1D serves as a loading control to show specific decline of CatSper1 and Caveolin1, which is in contrast with no change in CatSper3 and CatSpere. To better represent the results and avoid confusion, we have also replaced CatSper1 blot in Figure 1H from a repeat experiment. Regardless, we have toned down this sentence and directed other panels as well “The molecular weight of CatSper1, among all CatSper subunits, specifically declines during sperm capacitation (Chung et al., 2014; Figure 1B, D, H, I, K)”.

The reviewer suggests an interesting possibility regarding the stoichiometry of the CatSper channel. We speculate that the stoichiometry is not likely to change during capacitation for the following reasons. First, mature sperm lack ER and Golgi and there is no new synthesis or assembly of CatSper channel subunits. Second, CatSper channel complexes (comprised of at least 10 gene products per complex) are linearly arranged in the specialized nanodomains in the flagellar membrane, which would not be favorable for lateral movement of a subunit between complexes. Third, the stoichiometry of the CatSper channel might remain the same if capacitation induced the cleavage only but not degradation of the whole CatSper1. Our CatSper1 antibody used throughout this study detects the N-terminal region of CatSper1 (Figure 1E and G). Currently, there is no antibody recognizing C-terminal CatSper1 available to directly test this idea.

3) Figure 1B, the figure legend for this panel is incomplete. In its current form, it is hard for the readers to understand what are the treatments in the panel.

Thank you for pointing out this oversight. We have revised the legend to include more information.

“Mouse CatSper1 is an O-glycosylated protein. Apparent molecular weights of CatSper1 proteins were analyzed by immunoblotting. […] The dotted line indicates different exposure time of the same membrane.”

4) Figure 1B, in the treatment with O-Gly, the cleavage appears quite incomplete. Did the density of the lower band increase with longer treatment?

We agree that our O-glycosidase treatment did not completely *de-glycosylate* CatSper1 in Figure 1B. As shown in Figure 1—figure supplement 1C, sperm CatSper1 is prone to be *cleaved* even at 4C in a time-dependent manner when sperm membrane fraction is isolated. Thus, we performed dose-dependent deglycosylation reaction instead of increasing the reaction time. We found that indeed the density of the lower band increased as shown in Author response image 1; we tested 0.2X and 2X of O-glycosidase considering the ratio of sperm number to O-glycosidase unit applied in Figure 1B (1x10^6^ cells/20,000 unit) is 1X.

**Author response image 1. sa2fig1:** 

5) Figure 1G, the legend states that "ND-CatSper1 proteins remain largely unchanged under the same conditions (bottom)", but there seems to be significant change in the figure. Please show statistics from several blots.

As the reviewer suggested, we have updated Figure 1G with quantification and statistics from several blots and provide raw data as source data. We have also replaced the blot image of ND-CatSper1 to better represent our observation. We have revised the figure legend accordingly.

6) Figure 1—figure supplement 1A: The figure legend suggests that CatSper1 is not a phosphoprotein, yet only tyrosine phosphorylation was explored. Does PP1 dephosphorylate serine/threonine as well? If not, the text should be updated to indicate that CatSper1 is not tyrosine phosphorylated.

In fact, PP1 (protein phosphatase 1) is a serine/threonine phosphatase. Therefore, we explored both S/T and Y phosphorylation of CatSper1. No changes in either PP1 or PTP treatment support our conclusion that CatSper1 is not a phosphoprotein. We have revised the text to clarify this point (subsection “CatSper1 undergoes post-translational modifications during sperm development and maturation”).

7) Figure 1—figure supplement 1E: Please indicate how you determined that A23178 induced CatSper1 processing occurs via the same pathway as PKA. Perhaps the authors have demonstrated that H89 prevents A23178 induced CatSper1 degradation? Or inhibit PKA signaled degradation by loading sperm with BAPTA-AM? Alternatively, it is possible that these be two pathways are distinct? If this is not definitively known, the authors should clarify the schematic to indicate that these steps are speculated.

Thank you for your comments and great suggestions to further clarify the signaling cascades. In fact, we *did not think* that A231178 induces CatSper1 processing occurs via the same pathway as PKA. We thought that these two pathways are distinct and only indirectly connected. Our model was based on the following observations:

– Our previous study demonstrated that CatSper1-mediated Ca^2+^ signaling crosstalk with capacitation-associated PKA phosphorylation cascades. Blocking Ca^2+^ entry by genetic ablation of CatSper or chelating extracellular Ca^2+^ by BAPTA potentiated PKA activity and its downstream pY development only under capacitating conditions. Consistently, Ca^2+^ entry via A23187 or ionomycin partly blocked this enhancement (Chung et al., 2014, see Figure 4, Supplementary Figure 5B and C and Supplementary Figure 6B).

– In this study, we have shown that (1) even in non-capacitating sperm, Ca^2+^ influx is sufficient to induce CatSper1 processing (Figure 1I); (2) under capacitating conditions, PKA inhibition either by H89 or ST-Ht31 (Figure 1H and Figure 1—figure supplement 1D) accelerates CatSper1 processing.

As noted by the reviewer, the above experimental conditions did not directly test the effect of Ca^2+^ influx alone (such as in the continuous presence of PKA inhibitor) or PKA inhibition itself (in the presence of intracellular Ca^2+^ chelator). We thank the reviewer for these excellent questions to further clarify the signaling pathways!

We newly performed these experiments and have revealed that (1) PKA inhibition (H89) itself induces CatSper1 processing to some extent, but A23187 treatment accelerate CatSper1 processing even in the presence of H89 under non-capacitation conditions (new Figure 1—figure supplement 1E, upper); (2) suppressing a rise in cytoplasmic Ca^2+^ by BAPTA-AM, regardless of PKA activity, did not prevent CatSper1 processing (new Figure 1—figure supplement 1E, bottom). All these results corroborate our hypothesis in our schematic (now Figure 1—figure supplement 1F) that the protease activity is localized close to the inner leaflet of the plasma membrane (but outside of CatSper nanodomains); the local Ca^2+^ influx via CatSper channel, which normally accompanies other capacitation-associated membrane events (i.e., cholesterol efflux), is essential for the protease activity. This pathway can be indirectly modulated by PKA phosphorylation cascades such as regulation of the protease activity by phosphorylating PP1/PP2A the activity.

We have included these new data as Figure 1—figure supplement 1E and revised the legend accordingly. We have clarified the text (subsections “CatSper1 degradation involves Ca^2+^ and phosphorylation-dependent protease activity” and “CatSper1 as a molecular barcode for sperm maturation and transition in the female tract”) and modified the schematic to indicate PKA and PP1/PP2A regulation of protease X is our speculation.

8) The motivation for use of the Su9DsRed; Acr EGFP mice is never stated in the manuscript. The authors should state why these sperm were used in both the text, and figure labels should be also be updated (e.g. Figure 2D – to which proteins/structures are these fluorophores anchored?).

Thank you for identifying the areas that need modification to improve our manuscript. Su9DsRed2/Acr-EGFP mice (Hasuwa et al., 2010) generate sperm that have green acrosome (EGFP fused to proacrosin signal sequence under Acrosin3 promoter) and red mitochondria (DsRed2 fused to a mitochondrial import sequence). Therefore, the sperm head is green prior to the acrosome reaction, but GFP is lost after acrosome reaction. RFP (DsRed2) expression is seen in the midpiece of the sperm regardless of the acrosome state. The mouse line has been utilized by us and others to track the location of sperm in the female reproductive tract and probe the acrosome state in the field (Tokuhiro et al., 2012; Fujihara et al., 2013; Chung et al., 2014; Kiyozumi et al., 2020). We have stated the rationale in the main text (subsection “Sperm cells capacitated in vivo become heterogeneous along the female tract with distinct molecular characteristics”) and updated the labels in Figure 2D. There is space constraint in figure label; we added one color component per each panel and revised the legend.

9) Figure 2: This figure would be substantially enhanced by inclusion of a pie chart for in vivo capacitated sperm, similar to the chart included for in vitro capacitated sperm. Also, did you look at pY in the cleared organs?

We were not able to look at pY in the cleared organs because anti-phosphotyrosine antibody (4G10), which is a phosphorylation moiety antibody, cross-reacts with many other proteins and yield low signal-to-noise to detect sperm. Now we have included quantification of pY from in vivo capacitated sperm as Figure 2F (please see also our response to revision point #11). We have now provided raw data and pie charts as source data related to this figure.

We did not quantify in vivo capacitated sperm for CatSper1 from micro-dissection. Our CatSper1 antibody is suitable for direct and specific labeling of sperm by in situ imaging as CatSper1 is a sperm-specific protein. All our quantification from in situ imaging (Figure 4B) and ANN-detection and quantification of sperm for CatSper1 (Figure 6) are statistically significant and strongly support our conclusion. In situ imaging is more desirable when applicable since micro-dissection takes time and cross-contamination to a certain extent during handling is unavoidable.

10) When describing the images in Figure 3, the text (subsection “3D in situ molecular imaging of gametes in the female reproductive tract”, first paragraph) should state the protein/tissues labeled by each stain.

Good suggestion. We have updated the text with the requested information (subsections “3D in situ molecular imaging of gametes in the female reproductive tract” and “ANN-quantified CatSper1 signal reveals a molecular signature of successful sperm in situ”).

11) Have you quantified tyrosine phosphorylation of in vivo sperm? Perhaps the relative fluorescence of immune-stained sperm? The concluding statement to the in vivo imaging paragraph (subsection “Sperm cell that successfully reach the ampulla are CatSper1-intact and acrosome reacted”) suggests that you do. Perhaps the evidence could be explicitly stated and readers directed to a figure panel. Again, including a pie chart with this data would be of great value to the readers (in Figure 2E).

Thank you for your suggestion to improve our presentation of the data. Now we have added a bar graph to Figure 2F with the relative proportion of pY (-) vs. pY (+) sperm in these three regions. As we have done for in vitro capacitated sperm (Figure 2B and C), we have simply classified pY immunofluorescence as pY (-) vs. pY (+). The representative images in Figure 2E show our criteria for this classification. We have also included the raw data points (the number of sperm counted for pY immunofluorescence from each region) and pie charts from 9 micro-dissected tracts from 5 females as Figure 2—source data 1. As suggested, these raw data points providing the information on the individual-to-individual and tract-to-tract variations will be of great value to the scientific community. We have explicitly directed readers to the newly added figures for our conclusion statement (subsection “Sperm cells capacitated in vivo become heterogeneous along the female tract with distinct molecular characteristics”).

12) The methods describing development of ANN for these experiments is missing information such as the criteria used for choosing ANN? What quantitative evaluations were used in making these decisions? What percentage of the data were set aside for training?

In fact, we dedicated one Results subsection “Automatic detection of sperm in the voluminous female tract using artificial neural network” to describe the method of developing ANN and the criteria, along with Figure 5 and Figure 5—figure supplements 1 and 2. Specifically, the developed ANNs were chosen according to their sensitivity and specificity in detecting sperm cells and somatic nuclei, and the abundance (voxel occupancy) of noise (subsection “Artificial neural network (ANN) image processing”). We have now further highlighted these aspects in the Materials and methods section and directed to the corresponding quantitative graphic output (Figure 5—figure supplement 2C). The approach to generate training environments was addressed in Figure 5—figure supplement 1, Generation of training environments for ANN development. In total, 10^4^ environments were generated and utilized (100%) in ANN training. In this revision submission, we have now also provided additional materials related to ANN development such as ANN input feeding, training, construction of numerical objects coordinates arrays for supervised learning and ANN performance evaluation as source data.

13) Were the sperm in Figure 6 (used for ANN analysis) immunolabeled for CatSper1?

Yes, the sperm in Figure 6 were immunolabeled for CatSper1 as described in Figure 6D and F. We have clarified this by stating in the figure title and have also revised the text accordingly.

“Figure 6. ANN assessment of quadrilateral CatSper nanodomains and Δ fluorescent intensity in sperm population along the cleared female tract immunolabeled for CatSper1 conform to findings by other approaches used in this study.”

“We isolated the fluorescent signal from a proximal region of the principal piece close to the annulus where the immunolabeled CatSper1 signal is the most intense (Figure 6B).”

14) Figure 6H. Left panel, what do the delta values refer to? Fluorescence intensity? Right panel: delta fluorescence, what values were used for the basal measurements?

The delta values refer to the delta value of the fluorescent intensity. We have now clarified the figure legends. The baseline (zero) in the right panel of Figure 6H indicates homogenous angular distribution of the CatSper fluorescent signals with no quadrilateral arrangement. We have now added this description in the figure legend. In our revision submission, we have also provided the statistical evaluation presented in the Figure 6H right panel as “Figure 6H right panel statistical outputs summary table” as part of Figure 6—source data 2 “3D in situ analytical tools for CatSper quadrilateral structure”. We have placed custom scripts, digital masks for CatSper1 fluorescent signal quadrilateral delta values quantification together with text, and a snapshot and video description on how to operate them.

15) A role for calpain proteolysis in CatSper1 degradation with in vitro capacitation is intriguing, however, it is not clear from the writing that this is a hypothesis. If this is not a hypothesis and Calpain is known to mediate CatSper1 proteolysis, a citation should be provided. If not, the writing should be edited to clarify that this is a hypothesis.

We have clarified and now explicitly present this concept as a hypothesis (subsections “CatSper1 degradation involves Ca^2+^ and phosphorylation-dependent protease activity” and “CatSper1 as a molecular barcode for sperm maturation and transition in the female tract”). We previously showed that CatSper1 degradation is blocked by 26S proteasome inhibitor, MG-132 (Chung et al., 2014). In this study, we find that proteolysis is Ca^2+^ dependent and inhibited by all three classes of Calpain inhibitors. Yet we currently do not know the identity of the responsible protease(s).